# Querying Kernel Methods Suffices for Reconstructing their Training Data

**Daniel Barzilai**\*                                                                 *daniel.barzilai@weizmann.ac.il*
*Weizmann Institute of Science*

**Yuval Margalit**\*                                                                 *yuval.margalit@weizmann.ac.il*
*Weizmann Institute of Science*

**Eitan Gronich**                                                                     *eitan.gronich@weizmann.ac.il*
*Weizmann Institute of Science*

**Gilad Yehudai**                                                                     *gilad.yehudai1@gmail.com*
*Courant Institute of Mathematical Sciences, New York University*

**Meirav Galun**                                                                     *meirav.galun@weizmann.ac.il*
*Weizmann Institute of Science*

**Ronen Basri**                                                                     *ronen.basri@weizmann.ac.il*
*Weizmann Institute of Science*

**Reviewed on OpenReview:** *https://openreview.net/forum?id=qikuoGOTh2*

## Abstract

Over-parameterized models have raised concerns about their potential to memorize training data, even when achieving strong generalization. The privacy implications of such memorization are generally unclear, particularly in scenarios where only model outputs are accessible. We study this question in the context of kernel methods, and demonstrate both empirically and theoretically that querying kernel models at various points suffices to reconstruct their training data, even without access to model parameters. Our results hold for a range of kernel methods, including kernel regression, support vector machines, and kernel density estimation. Our hope is that this work can shed light on potential privacy concerns associated with such models.

## 1 Introduction

Machine learning methods often rely on highly expressive models for performing well on various tasks (Zhang et al., 2021; Kaplan et al., 2020). However, this success comes at a cost: these models often memorize large parts of their training data, raising significant concerns about unintended privacy leaks (Carlini et al., 2023a; Haim et al., 2022). As a result, understanding memorization has become a central subject of research in the past few years due to its importance, both theoretically and practically. Recent theoretical works have suggested that memorizing a constant fraction of the training data may be inevitable in certain settings (Brown et al., 2021; Attias et al., 2024). However, it is still unclear when memorization translates into privacy vulnerabilities, especially in scenarios where attackers have only limited (e.g., query-only) access to the model.

Kernel methods are a popular set of tools that offer an ideal proving ground for this question. In particular, kernels are both highly expressive (capable of severe memorization) and analytically tractable. This allows us to study fundamental questions about memorization in settings that reflect key characteristics of modern

---

\*Equal Contribution

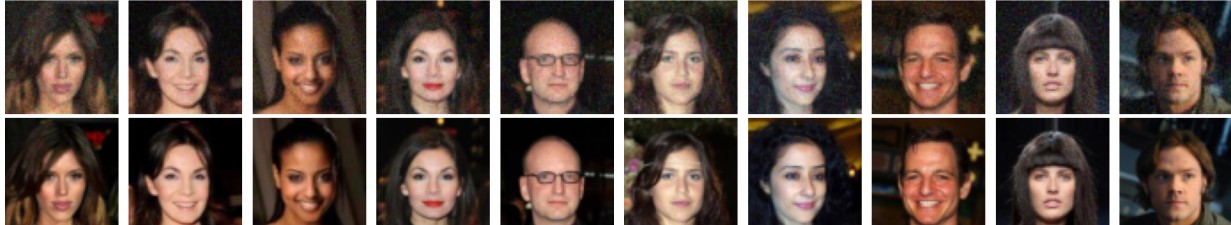

Figure 1: Reconstruction of training images in a kernel regression task with an RBF kernel pre-trained on 500 images from the celebA dataset. The top row shows 10 reconstructions, and the bottom row shows their nearest neighbors in the training set. The full set of reconstructions can be found in the appendix in Fig. 8.

learning systems (e.g., query-only settings). In particular, we will be interested in the following question: *Given query-only access to a kernel model, can we reconstruct its training data?*

In this work, we both prove theoretically and demonstrate empirically that the answer to the above question is *yes*: The ability to query models at multiple points is sufficient to mount an effective data reconstruction attack. We show that in this setting, data reconstruction attacks pose a major security concern, even without access to model parameters. We summarize our key contributions as follows.

1. We present a data reconstruction attack that works in settings where the attacker only has access to model queries but not to model parameters. This attack is applicable for a variety of classical learning algorithms, including kernel regression, kernel support vector machines, and kernel density estimation.

2. On the theoretical side, we prove that for a wide range of kernels, minimizing our reconstruction loss guarantees reconstructing the training set. Furthermore, we formally prove an upper bound on the number of query points needed in order to gather enough information on the attacked model to reconstruct its entire training set.

3. We demonstrate our reconstruction attack empirically in a range of settings, in many cases recovering the majority of the training set. While there are no works to directly compare against, we show that the quality of our reconstructions is better than or comparable to reconstruction attacks in other settings (even ones that access model parameters).

Fig. 1 shows examples of celebrity images reconstructed from a trained RBF kernel. These results challenge the assumption that preventing access to model parameters mitigates privacy risks. By exposing vulnerabilities across various settings, we hope our work highlights the need for privacy-preserving techniques that remain robust even in black-box settings.

## 2 Related Works

**Data Reconstruction.** Extracting sensitive information from trained models is the subject of many studies. Early works include *model inversion attacks* (Fredrikson et al., 2015; He et al., 2019; Yang et al., 2019), also known as activation maximization, that aim at reconstructing class representatives by optimizing over the inputs to maximize the desired class output. Although there are semantic similarities between the reconstructions and the trained data, these are still not the true data samples used to train the attacked model. Recently, Haim et al. (2022) demonstrated a reconstruction attack based on theoretical results on the implicit bias of neural networks (Lyu & Li, 2019; Ji & Telgarsky, 2020). This work was later extended and further analyzed (Buzaglo et al., 2024; Oz et al., 2024; Smorodinsky et al., 2024), and was adapted to the lazy regime (Loo et al., 2023). We emphasize that all the above methods are not directly comparable to ours, as their methods require working with the parameters of the trained model, which for kernel methods is usually intractable. Our method only requires query access to the evaluations of the model on new points. Other methods demonstrated the reconstruction of training data in large language (Carlini et al., 2021; 2022)

and diffusion models (Carlini et al., 2023b; Somepalli et al., 2023). These methods are specifically designed for generative models.

Notably, (Tramèr et al., 2016) empirically demonstrated potential reconstruction attacks against a wide range of methods in a query-only fashion. However, their results for kernel methods were on a very small scale and served primarily as a proof of concept. In particular, they reconstructed at most 20 MNIST-style images of resolution $14 \times 14$ from an RBF kernel trained on the logistic loss. Moreover, we provide theoretical grounding for our reconstruction attack as well as an understanding of why such a query-based reconstruction attack works. Lastly, we introduce many algorithmic and methodological differences, including leveraging synthetic data, optimizing unknown hyperparameters, and performing dimensionality reduction.

**Kernel Methods.** Kernel methods are a popular set of tools for solving various tasks, including kernel regression, Support Vector Machines (SVM), and kernel density estimation (James et al., 2013; Devroye et al., 2013). In recent years, kernel methods have seen renewed popularity due to their connection to overparameterized neural networks (Lee et al., 2017; Jacot et al., 2018; Arora et al., 2019; Lee et al., 2019; Chizat et al., 2019; Du et al., 2019; Allen-Zhu et al., 2019; Yang & Littwin, 2021). While this connection is only approximate and does not encompass all relevant settings, kernel methods have proved to be a valuable tool in understanding intriguing phenomena in neural networks, including, for example, double/multiple descent (Belkin et al., 2019; Mei et al., 2022; Xiao & Pennington, 2022; Barzilai & Shamir, 2024), frequency bias (Bietti & Mairal, 2019; Basri et al., 2020; Barzilai et al., 2023), and benign overfitting (Hastie et al., 2022; Tsigler & Bartlett, 2023). From a practical perspective, kernel methods remain common tools, as they may outperform neural networks in small data or low-dimensional tasks (Arora et al., 2020).

## 3 Preliminaries on Kernel Methods

Kernel methods learn a linear function in some feature map $\phi : \mathcal{X} \to \mathcal{H}$, where $\mathcal{X} \subseteq \mathbb{R}^d$ and $\mathcal{H}$ is a Hilbert space with norm $\|\cdot\|$. Consider a dataset of $N$ inputs that are $d$-dimensional $\mathbf{x}_1, \ldots, \mathbf{x}_N \in \mathbb{R}^d$ and possibly $C$-dimensional labels $\mathbf{y}_1, \ldots, \mathbf{y}_N \in \mathbb{R}^C$. Then, for parameters $\mathbf{w}_1, \ldots, \mathbf{w}_C \in \mathcal{H}$, kernel methods learn a function of the form

$$f(\mathbf{x}) = [\langle \mathbf{w}_1, \phi(\mathbf{x}) \rangle, \ldots, \langle \mathbf{w}_C, \phi(\mathbf{x}) \rangle]^\top \in \mathbb{R}^C. \tag{1}$$

Let $\boldsymbol{k} : \mathcal{X} \times \mathcal{X} \to \mathbb{R}$ be the kernel defined as $\boldsymbol{k}(\mathbf{x}, \mathbf{x}') := \langle \phi(\mathbf{x}), \phi(\mathbf{x}') \rangle$. A useful observation that is central to our reconstruction attack is that for many learning algorithms, the learned predictor can be expressed using the kernel function as

$$\forall c \in [C], \ f_c(\mathbf{x}) = \sum_{i=1}^{N} \alpha_{i,c} \boldsymbol{k}(\mathbf{x}, \mathbf{x}_i), \qquad \text{for some } \alpha_{i,c} \in \mathbb{R}. \tag{2}$$

We detail two scenarios in which $f$ takes a form as in Eq. (2). The first is given by the Representer theorem (Schölkopf et al., 2001; Micchelli & Pontil, 2005) which states that any $f$ that minimizes a loss function that also includes a regularization term on the norms of the parameters can be written as in Eq. (2).

The second scenario involves training by gradient-based methods. Consider the case where the weights $\mathbf{w}_i$ are initialized at zero and trained by running a gradient-based optimization method over a loss function $\ell$. Then it is straightforward to observe that the gradients $\nabla_{\mathbf{w}_i} \ell(f(\mathbf{x}_1), \mathbf{y}_1, ..., f(\mathbf{x}_N), \mathbf{y}_N)$ must lie in the span of $\phi(\mathbf{x}_i)$. As a result, for many variants of gradient descent, the parameters $\mathbf{w}_i$ remain in the span of $\phi(\mathbf{x}_i)$ throughout training. It is straightforward to verify that when this occurs, the learned predictor $f$ can be represented as in Eq. (2).

Importantly, the function $f$ is characterized by the kernel $\boldsymbol{k}$, and therefore does not require explicit computation of the feature map $\phi$ (often referred to as the kernel trick). This allows considering $\phi$ that may be infinite-dimensional. As is common, we will often describe kernels through $\boldsymbol{k}$ without describing the feature map $\phi$ explicitly.

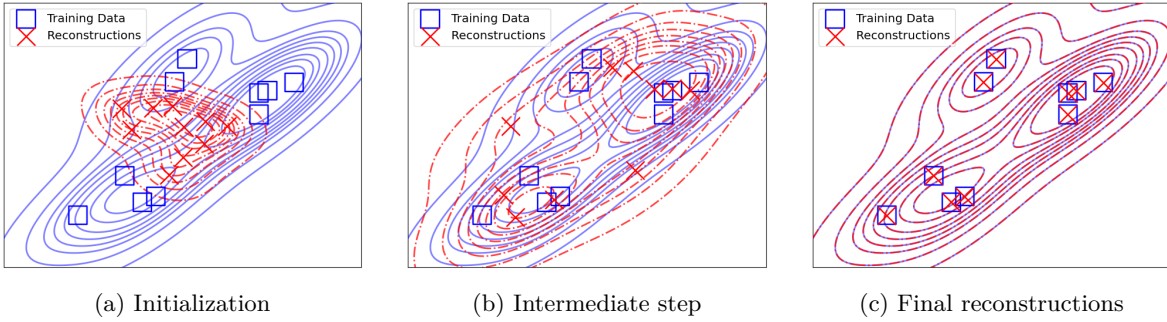

(a) Initialization                (b) Intermediate step                (c) Final reconstructions

Figure 2: Reconstruction of training points from a two-dimensional kernel density estimator that was trained on the ground truth data points marked by blue squares. We initialize our reconstruction with random samples (left panel). The blue contours represent the attacked density estimator $f$. The red dashed contours represent the model generated by our reconstructions at different steps of optimizing Eq. (4). The reconstructed points (marked by red crosses) match the ground truth points at convergence (right panel).

---

**Algorithm 1** Query-Based Kernel Reconstruction Attack (fine details in Appendix A)

---

**Input:** Attacked model $f$, kernel $\boldsymbol{k}$, number of points to reconstruct $n$, query distribution $\mathcal{D}$, iterations $T$
**Output:** reconstructed points $\{\hat{\mathbf{x}}_i\}_{i=1}^n$ and coefficients $\{\hat{\alpha}_{i,c}\}$

1: Initialize $\hat{\mathbf{x}}_i$ and $\hat{\alpha}_{i,c}$ randomly for $i \in [n]$, $c \in [C]$
2: Sample query points $\{\mathbf{z}_j\}_{j=1}^m \sim \mathcal{D}$ i.i.d.
3: **for** $t = 1$ **to** $T$ **do**
4:    Compute loss $\mathcal{L}_{\text{rec}}(P) = \frac{1}{mC} \sum_{j=1}^m \sum_{c=1}^C \left( \sum_{i=1}^n \hat{\alpha}_{i,c} \boldsymbol{k}(\mathbf{z}_j, \hat{\mathbf{x}}_i) - f_c(\mathbf{z}_j) \right)^2$
5:    Update $\hat{\mathbf{x}}_i$, $\hat{\alpha}_i$ via an optimization step
6: **end for**
7: **return** $\{\hat{\mathbf{x}}_i\}, \{\hat{\alpha}_{i,c}\}$

---

## 4 Query-Based Reconstruction Attack

We now detail the reconstruction attack, which is summarized in Algorithm 1. Suppose that a model $f$ that was trained on a dataset $\mathbf{x}, \ldots, \mathbf{x}_N$ is expressed as in Eq. (2). We call $f$ the attacked model. Let $n$ be the number of samples we aim to reconstruct, and define the reconstruction parameters that we optimize

$$P := \{\hat{\alpha}_{i,c}\}_{i \in [n], c \in [C]} \bigcup \{\hat{\mathbf{x}}_i\}_{i \in [n]}. \tag{3}$$

We choose a query distribution $\mathcal{D}$, then sample $m$ query points $\mathbf{z}_1, \ldots, \mathbf{z}_m \sim \mathcal{D}$ (more on the choice of $\mathcal{D}$ below), and optimize the parameters $P$ using the following reconstruction loss:

$$\mathcal{L}_{\text{rec}}(P) := \frac{1}{mC} \sum_{j=1}^m \sum_{c=1}^C \left( \sum_{i=1}^n \hat{\alpha}_{i,c} \boldsymbol{k}(\mathbf{z}_j, \hat{\mathbf{x}}_i) - f_c(\mathbf{z}_j) \right)^2 \tag{4}$$

The reconstruction loss gives $m \cdot C$ non-linear equations that the parameters $P$ should satisfy.

Importantly, our reconstruction attack accesses the attacked model $f$ only through its evaluation of new points and does not require access to its parameters $\mathbf{w}_i$. Access to model parameters does not have to be made public, even if the underlying architecture (in our case, the feature map $\phi$) is publicly known. Furthermore, our attack does not require explicitly working in the feature space. Many feature maps that we consider are very high-dimensional (or even infinite-dimensional), and thus, reconstruction attacks that work in feature space may be computationally expensive and perhaps even completely infeasible.

### 4.1 Threat Model

We now detail exactly what the attacker may access/know, and what is unavailable to the attacker.

**Query access.** We assume the attacker has unlimited access to evaluate the attacked model $f$ at arbitrary points. However, the attacker may not access the parameters of the model $f$.

**Kernel type and hyperparameters.** The attack requires knowledge of the kernel $k$ used by the attacked model $f$, including the kernel hyperparameters. We emphasize however that this can be relaxed in several ways, as we later discuss in Sec. 6 and demonstrate in Table 4. In many cases, various hyperparameters can be learned by the attacker together with the parameters $P$. As such, the attack may still be effective when only the *family* of kernels is known. From a practical perspective, we note that there are a few common families of kernels, and an attacker may attempt the reconstruction attack with each of these.

**Underlying distribution.** An important aspect of this reconstruction attack is that the requirements on the queries are relatively mild. First, the query points $\mathbf{z}_j$ do not require labels. Second, the query distribution $\mathcal{D}$ does not need to be identical to the training distribution of $f$, and indeed the theoretical analysis shows that $\mathcal{D}$ can be any distribution with a density. For practical purposes, it is sensible to choose $\mathcal{D}$ to be of the same modality as the training distribution, e.g. a large public image dataset if the training samples $\mathbf{x}_i$ are natural images. Throughout our experiments, we will often use synthetic data for the query points (see 7 for more details).

**Number of points to be reconstructed.** The attack does not require exact knowledge of the number of training points $N$. As we will later demonstrate in Table 3, the attack works when the number of samples to be reconstructed $n$ is either larger or smaller than the number of training points. However, the attack works best when $n \geq N$, and therefore, if the number of training points $N$ is unknown, it is better to over-guess.

### 4.2 Use Cases

Throughout this paper, we consider attacked models $f$ that arise from various training algorithms and kernels $k$. For the choice of kernel, we consider choices that are common in the literature, including the Laplace kernel, the Gaussian (RBF) kernel, the NTK, and polynomial kernels (see Appendix C for definitions). We now provide several concrete settings for which the reconstruction attack is applicable.

**Kernel Regression (KRR).** Here, the parameters $\mathbf{w}_1, \ldots, \mathbf{w}_C$ of $f$ are chosen to minimize the regularized mean-squared error loss

$$\frac{1}{2N} \sum_{i=1}^{N} \sum_{c=1}^{C} \left( \langle \mathbf{w}_c, \phi(\mathbf{x}_i) \rangle - y_{i,c} \right)^2 + \frac{\lambda}{2} \|\mathbf{w}_c\|^2 \tag{5}$$

with $\lambda \geq 0$. It is well known that letting $K \in \mathbb{R}^{N \times N}$ be the kernel matrix given by $K_{ij} = k(\mathbf{x}_i, \mathbf{x}_j)$ and $Y := (\mathbf{y}_1, \ldots, \mathbf{y}_N)^\top \in \mathbb{R}^{N \times C}$, the minimizer of Eq. (5) is given by a function $f$ satisfying Eq. (2) with

$$\alpha_{i,c} = \left( (K + N\lambda I)^{-1} Y \right)_{i,c} \quad \forall i \in [N], \ c \in [C].$$

Unless stated otherwise, we take $\lambda = 0$. We find that the reconstruction attack is relatively insensitive to the choice of $\lambda$. Examples of reconstructions with $\lambda > 0$ can be found in Appendices 14 and 15.

**Support Vector Machines (SVM).** For (hard) SVM, we consider the case where $f$ is trained to minimize the hinge loss, which for binary labels $y_i \in \{\pm 1\}$ is given by $\frac{1}{N} \sum_{i=1}^{N} \max(0, 1 - f(\mathbf{x}_i)y_i)$. For multi-class classification, letting $y_i \in \{1, \ldots, C\}$ be the class labels, the hinge loss is defined as (Crammer & Singer, 2001)

$$\frac{1}{N} \sum_{i=1}^{N} \max \left( 0, 1 + \max_{c \neq y_i} \langle \mathbf{w}_c, \phi(\mathbf{x}_i) \rangle - \langle \mathbf{w}_{y_i}, \phi(\mathbf{x}_i) \rangle \right),$$

where $\mathbf{w}_1, \ldots, \mathbf{w}_C$ are the parameters of $f$. Unlike kernel regression, there is no closed-form solution for the minimizer of this loss. Nevertheless, we may consider an attacked model $f$ that was trained on this loss by gradient descent. As mentioned in Sec. 3, training by gradient descent ensures that $f$ indeed is of the form Eq. (2). Due to the high-dimensionality of $\phi$, we express $f$ as in Eq. (2) and train directly on $\alpha_{i,c}$ so that we never have to compute $\phi$ and $\mathbf{w}$ directly.

**Kernel Density Estimation (KDE).** Here, the task is to approximate a density function of a target distribution given only a finite number of samples. Specifically, given some points $\mathbf{x}_1, \ldots, \mathbf{x}_n \in \mathbb{R}^d$ drawn from a distribution with unknown probability density function (PDF) $p$, kernel density estimation (KDE) is a well-known method to learn a function $f$ that estimates $p$. We next show how querying the function $f$ can leak the entire training data.

Consider the normalized Gaussian kernel given by

$$\boldsymbol{k}_H(\mathbf{x}, \mathbf{x}') = \frac{1}{\sqrt{(2\pi)^d |H|}} \exp\left(-\frac{1}{2}\left\| H^{-\frac{1}{2}}(\mathbf{x} - \mathbf{x}') \right\|_2^2\right) \tag{6}$$

for a matrix $H \succ 0$. Then, KDE estimates $p$ using $f(\mathbf{x}) := n^{-1} \sum_{i=1}^N \boldsymbol{k}_H(\mathbf{x}, \mathbf{x}_i)$. Clearly, $f$ is a valid PDF since each $\boldsymbol{k}_H(\mathbf{x}, \mathbf{x}_i)$ is a Gaussian when viewing $\mathbf{x}_i$ as fixed. It is well known that the estimator $f$ converges to the true density function $p$ as the number of samples grows (Devroye et al., 2013). Furthermore, $f$ fits our framework as it can be written in the form given by Eq. (2) with $C = 1$ and $\alpha_i = 1/N$, $\forall i \in [N]$.

We next provide an illustrative example of our attack using a 2-dimensional density estimation task. Suppose the underlying PDF $p$ is given by a mixture of two Gaussians, specifically $p(\mathbf{x}) = \frac{1}{2}\mathcal{N}(-\mu, I_2) + \frac{1}{2}\mathcal{N}(\mu, I_2)$, $\mu = (2, 2)^T$. A KDE model $f$ is obtained by sampling $N = 10$ points from $p$, $\{\mathbf{x}_i\}_{i=1}^{10}$, and computing an estimator $f(\mathbf{x}) := N^{-1} \sum_{i=1}^N \boldsymbol{k}_H(\mathbf{x}, \mathbf{x}_i)$ with $\boldsymbol{k}_H$ as in Eq. (6). For this example, suppose the attacked model $f$ picks $H$ according to what is known as Scott's rule (Scott, 2015)[Chapter 6], i.e., $H$ is diagonal with $H_{jj} = n^{-1/6}\tilde{\sigma}_j$, where $\tilde{\sigma}_j$ is the empirical standard deviation of $\{\mathbf{x}_i\}_{i=1}^N$ in the $j^{\text{th}}$ coordinate.

We visualize our reconstruction attack in Fig. 2. Given the attacked model $f$ as described above, our goal is to optimize the parameters $P$ (See Eq. (3)). Here, we do not assume that $H$ is known and also learn an estimate of $H$ that we denote by $\hat{H}$. $P$ and $\hat{H}$ are optimized by sampling query points $\mathbf{z}_j$ from a grid and minimizing the reconstruction loss given by Eq. (4). Each step of the optimization produces an approximation to the attacked model of the form

$$\hat{f}(\mathbf{x}) := \sum_{i=1}^n \hat{\alpha}_i \boldsymbol{k}_{\hat{H}}(\mathbf{x}, \hat{\mathbf{x}}_i) \tag{7}$$

that becomes very close to the attacked model $f$ as the reconstruction loss Eq. (4) approaches 0. As can be vividly seen in Fig. 2, when $\hat{f}$ approaches $f$, the reconstructed training points $\hat{\mathbf{x}}_i$ approach the true training points $\mathbf{x}_i$. We will show in Sec. 5 that this is not by coincidence and is a necessary condition for the loss to be minimized.

## 5 Theoretical Guarantees

In this section, we prove theoretically the effectiveness of the proposed reconstruction attack. Our result will be stated for kernels which are *strictly p.d.*, meaning that for every set of distinct points $\mathbf{x}_1, ..., \mathbf{x}_n \in \mathcal{X}$ the kernel matrix $K_{ij} = \boldsymbol{k}(x_i, x_j)$ is strictly positive definite. Kernels used in practice tend to be p.d., as many algorithms break if the kernel matrix is not well-conditioned . It is well known that many common kernels satisfy this property, such as the NTK (Carvalho et al., 2025) and bounded translation invariant kernels on $\mathbb{R}^d$ (Sriperumbudur et al., 2011). This includes the Laplace and RBF kernels that we use in this paper.

We will require that the kernel be analytic, except for possibly isolated singularity points, such as in the Laplace kernel. Formally, we define the following mild condition, which is satisfied by all kernels we consider in this paper.

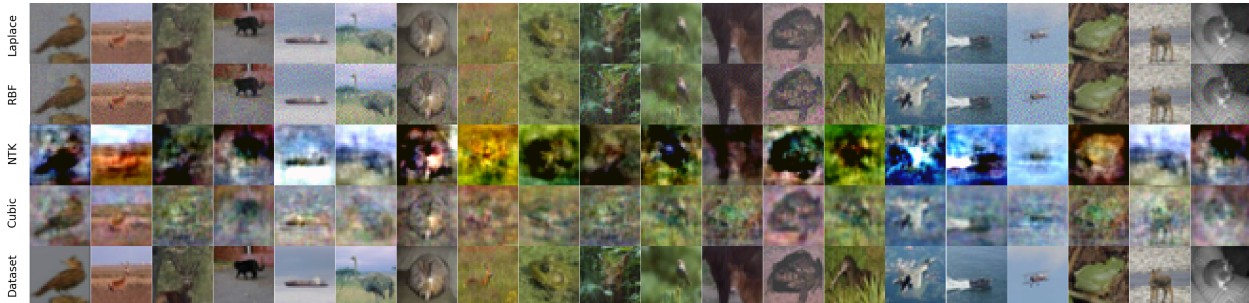

Figure 3: Top reconstructions from multiple kernel models trained on CIFAR10 images. Bottom row: training images from the dataset. Rows 1-4: reconstructions that are nearest to the bottom row images, obtained by attacking trained Laplace kernel, RBF kernel, cubic polynomial kernel, and NTK, respectively.

**Definition 1.** We say a kernel is *almost analytic* if there exists a countable family of $C^1(\mathcal{X})$ functions $\{\Gamma_s : \mathcal{X} \to \mathcal{X}\}_{s \in \mathbb{N}}$ such that for any $(\mathbf{x}, \mathbf{z}) \in \mathcal{X} \times \mathcal{X}$, if $\mathbf{z} \notin \bigcup_{s \in \mathbb{N}}\{\Gamma_s(\mathbf{x})\}$ then there is a neighborhood around $(\mathbf{x}, \mathbf{z})$ on which $\boldsymbol{k}$ is analytic.

Note that all common kernels, including RBFs, Laplace, NTK, and polynomials, are almost analytic. This assumption is made solely to exclude pathological cases in which the value of the attacked model on infinitely many random points does not determine the function globally. For example, non-analytic functions can be constant over large regions, making precise reconstruction from their values impossible.

We are now ready to state our main theorem, which ensures that given sufficiently many query points, reconstruction of both the attacked function as well as the underlying data used to train it.

**Theorem 2.** *Let $\mathcal{X} \subseteq \mathbb{R}^d$ be open and $\boldsymbol{k} : \mathcal{X} \times \mathcal{X} \to \mathbb{R}$ be strictly p.d. and almost analytic. Let $\mathcal{D}$ be any distribution given by a density over $\mathcal{X}$. Let $f$ be an attacked predictor as in Eq. (2), where the data $\{\mathbf{x}_i\}_{i=1}^N$ are distinct and $\boldsymbol{\alpha}_i \neq \mathbf{0}$. Let $n \geq N$, $m > n(d+2)$, and let $\hat{\alpha}_{i,c}, \hat{\mathbf{x}}_i$ be any solution to the minimization problem defined by the reconstruction loss in Eq. (4), then it holds with probability 1 over $\mathbf{z}_1, ..., \mathbf{z}_m \sim \mathcal{D}^m$ that $f_c(\mathbf{v}) = \hat{f}_c(\mathbf{v}) := \sum_{i=1}^n \hat{\alpha}_{i,c}\boldsymbol{k}(\mathbf{v}, \hat{\mathbf{x}}_i)$ for all $\mathbf{v} \in \mathcal{X}$, $c \in [C]$, and*

$$\forall\, i \in [N] \quad , \quad \exists\, j \in [n] \qquad s.t \qquad \hat{\mathbf{x}}_j = \mathbf{x}_i.$$

A few notes are in order. In the above theorem, $\mathcal{X}$ is an open subset of $\mathbb{R}^d$, but see Remark 7 for a generalization to smooth kernels on submanifolds of dimension $d$ (such as $\mathbb{S}^d$). Furthermore, the distribution $\mathcal{D}$ must be defined by a density, and in particular, we use the property that $\mathcal{D}(E) = 0$ for every $E \subseteq \mathcal{X}$ with 0 (Lebesgue) volume.

An important property of Thm. 2 is that the size of the optimized training set $n$ can be chosen as an upper bound on the number of training points $N$ of the attacked model (which may be unknown). The result $\hat{\mathbf{x}}_i, \hat{\alpha}_{i,c}$ of the optimization would then contain either repeated instances of a training point $\hat{\mathbf{x}}_{i_1} = ... = \hat{\mathbf{x}}_{i_t} = \mathbf{x}_i$ (with $\forall c \in [C] : \alpha_{i,c} = \sum_{j=1}^t \hat{\alpha}_{i_j,c}$) or meaningless points $\hat{\mathbf{x}}_i$ with zero coefficients $\forall c \in [C] : \hat{\alpha}_{i,c} = 0$.

The full proof is found in Appendix B.2. The theorem in fact follows from the case $C = 1$, which can then be applied $\forall c \in [C]$. The first step is to show that w.p. 1 over $Z := (\mathbf{z}_1, ..., \mathbf{z}_m) \sim \mathcal{D}^m$, every predictor $\hat{f}(\mathbf{z}) = \sum_{i=1}^n \hat{\alpha}_i \boldsymbol{k}(\mathbf{z}, \hat{\mathbf{x}}_i)$ satisfying $\forall j \in [m] : f(\mathbf{z}_j) = \hat{f}(\mathbf{z}_j)$ actually satisfies $\forall \mathbf{z} \in \mathcal{X} : \hat{f}(\mathbf{z}) = f(\mathbf{z})$. As we show in the appendix in Proposition 3, this implies that the training data of the minimizer $\hat{f}$ and the true predictor $f$ are identical.

The main tool in proving that $\hat{f}(\mathbf{z}) = f(\mathbf{z})$ everywhere is the Submersion Level Set Theorem (see for instance, Lee (2012)). The theorem allows us to prove that when $m > n(d+2)$, the set of $m$-tuples of queries $Z = (\mathbf{z}_1, ..., \mathbf{z}_m)$ for which there exists a predictor $\hat{f} \not\equiv f$ of the above form satisfying $\forall j \in [m] : f(\mathbf{z}_j) = \hat{f}(\mathbf{z}_j)$, is contained in a projected manifold structure of dimension smaller than $md$, hence comprising a null set in $\mathcal{X}^m$. The challenge with this proof strategy is ensuring that each query $\mathbf{z}_j$ contributes a non-degenerate constraint on the training set, even when gradients of the predictor $\nabla_z \hat{f}(\mathbf{z}_j)$ vanish; this is guaranteed by

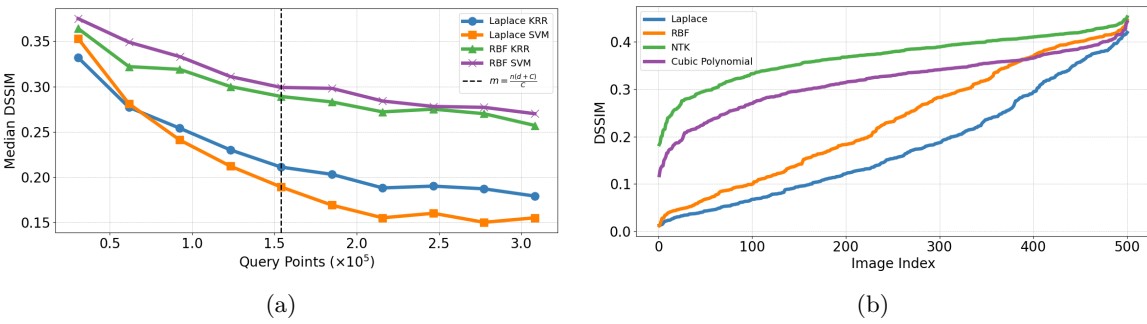

(a)                                                                  (b)

Figure 4: Reconstruction quality for different kernels and use cases on CIFAR10. Left: Comparison between kernel regression (KRR) and SVM with the Laplace and RBF kernels. Each point in the graph shows the median DSSIM obtained in a different run with a different number of query points. Right: Comparison between different kernels trained using KRR. In this figure, each graph shows a cumulative plot from one run, showing the quality of reconstruction, measured with DSSIM, obtained with the best $k$ images for $k \in [n]$.

the analyticity condition on $\boldsymbol{k}$, which implies that at least one higher order derivative of $\hat{f}$ does not vanish where $\hat{f}(\mathbf{z}) = 0$ (otherwise $\hat{f} \equiv 0$). For kernels $\boldsymbol{k}$ which are analytic everywhere, $m > n(d+1)$ queries suffice; the additional $n$ queries are used in the proof to ensure that w.p. 1 over $(\mathbf{z}_1, ..., \mathbf{z}_m)$, there is no predictor which has less than $n(d+1)$ analytic points among $(\mathbf{z}_1, ..., \mathbf{z}_m)$.

Given that the above set of tuples for which uniqueness of the predictor does not hold is a null set in $\mathcal{X}^m$, we have the desired result, namely that w.p. 1 on $Z \sim \mathcal{D}^m$, no such $Z$ is sampled, hence no such $\hat{f}$ exists for $Z$.

Note that when $C > 1$, the training points $\mathbf{x}_i$ are shared by $f_1, ..., f_C$, a property not utilized by Thm. 2 which could potentially reduce the number of necessary queries. In this case, each query $\mathbf{z}_j$ contributes $C$ constraints and the total number of parameters is $n(d+C)$, hinting that a bound closer to $m > \frac{n(d+C)}{C}$ queries should suffice to guarantee the usage of the Submersion Level Set Theorem. For this, we would need the predictors $f_1, ..., f_C$ to be sufficiently "different". Specifically for every set of predictors $\hat{f}_1, ... \hat{f}_C$ defined by linearly independent coefficient vectors $\hat{\boldsymbol{\alpha}}_{:,1}, ... \hat{\boldsymbol{\alpha}}_{:,C} \in \mathbb{R}^n$, the gradients of $\hat{f}_1, ..., \hat{f}_C$ would need to be linearly independent at points $\mathbf{z}$ where $\forall c \in [C] : \hat{f}_c(\mathbf{z}) = 0$. Note that these gradients are a set of $C$ vectors in $\mathbb{R}^d$ (and typically $C \ll d$), so this is likely to occur in practice when running Algorithm 1.

## 6    Analysis

**Reconstruction Without Kernel Knowledge.**    The reconstruction attack in this paper assumes knowledge of the kernel $\boldsymbol{k}$ used by the attacked model $f$. Nevertheless, many kernels share a very common structure and differ only by a few hyperparameters. It is therefore possible to optimize for these hyperparameters and thus reconstruct the training data without exact knowledge of the kernel. For example, consider the family of Mattern kernels, given by $\boldsymbol{k}(\mathbf{x}, \mathbf{x}') = \exp\left(-\gamma \|\mathbf{x} - \mathbf{x}'\|_2^{\beta}\right)$. This family encompasses Laplace kernels (with $\beta = 1$) and RBF kernels (with $\beta = 2$). In Table 4 we run an experiment for reconstruction where the values of $\beta$ and $\gamma$ of the attacked model are unknown. Specifically, we initialize $\hat{\beta} = 0.15, \hat{\gamma} = 0.01$, and run our attack as before using the reconstruction loss in Eq. (4). Unlike before, we also compute the derivatives of the loss with respect to $\hat{\beta}$ and $\hat{\gamma}$ and optimize them as well. We observe that $\hat{\beta}$ and $\hat{\gamma}$ converge near the real unknown parameters $\beta$ and $\gamma$ of the attacked model. We also observe that the quality of the reconstructions not only remains high, but in some cases may even surpass the quality when $\beta$ and $\gamma$ are known.

**Comparison Between Different Kernels.**    We find that the choice of kernel affects the effectiveness of our attack. Fig. 4 shows a quantitative evaluation of our results with different tasks (attacking KRR and SVM models) and with different choices of kernels (Laplace, RBF, NTK, and cubic polynomial). It appears

that attacking the Laplace kernel is more effective than the RBF kernel, and both are more effective than cubic and NTK models. There is no clear difference between attacking KRR and SVM models.

We infer that these differences in reconstruction quality across kernels is due to the different reconstruction loss landscapes that they induce, making some easier to optimize than others. Consider for example a Laplace kernel given by $\boldsymbol{k}(\mathbf{x}, \mathbf{x}') = \exp\left(-\gamma \left\|\mathbf{x} - \mathbf{x}'\right\|_2\right)$ for some $\gamma > 0$. As $\gamma \to 0$, the kernel is close to a constant function, and as $\gamma \to \infty$, the kernel approaches a delta function. In both extremes, one can expect reconstruction to be numerically infeasible. We verify this intuition empirically in Fig. 7 where we plot the reconstruction quality for different values of $\gamma$ ranging from 0.01 to 0.3. We observe a U shape when plotting the median reconstruction quality measured by DSSIM (see Sec. 7 for details), whereby the DSSIM worsens when $\gamma$ is too large or too small. For reference, Haim et al. (2022) considered reconstructions with a DSSIM less than 0.3 as high-quality. In any case, extreme values of $\gamma$ are less useful for regression and classification tasks and are, therefore, unlikely to be used in practice.

**Number of Query Points Needed.** Since our attack requires sampling $m$ query points $\mathbf{z}_i$, one may wonder how many query points are needed to obtain good reconstructions. Thm. 2 gives a strict bound that $m > n(d+2)$ is sufficient. However, the discussion following the theorem as well as the empirical experiments suggest that in practice, even fewer query points may suffice, closer to the order of $\frac{n(d+C)}{C}$. Practically, we observe (see Fig. 4) that there is no hard threshold for $m$ under which reconstructing the training set is impossible. Instead, we observe a relatively steady improvement as $m$ increases.

Importantly, because the complexity of our reconstruction attack does not depend on any parameter count, even when $m$ is very large, the attack may still be efficient relative to attacks that access parameters. For example, Haim et al. (2022); Buzaglo et al. (2024) optimize the same number of parameters as us, but the running time needed to compute the loss function scales as $p \cdot n \cdot C$ where $p$ is the number of parameters. In the 3-layer network Haim et al. (2022) considered, $p = 10^6 \cdot d$. In contrast, the running time complexity of our reconstruction attack scales with $m$ and the complexity of computing the kernel $\boldsymbol{k}$ instead of $p$. Thus, even when using many query points, our attack may still be more efficient.

**Dimensionality Reduction.** The number of query points needed to reconstruct the training data depends on the dimension of the reconstructed points. It is, therefore, natural to consider reducing the dimension of $\hat{\mathbf{x}}_i$ to decrease the number of query points needed. Specifically, consider a function $\Psi : \mathbb{R}^k \to \mathbb{R}^d$ that "decodes" or "upscale" vectors of dimension $k \ll d$ into vectors of dimension $d$. One may thus let $\hat{\mathbf{x}}_i = \Psi(\hat{\mathbf{v}}_i)$ for some vectors $\hat{\mathbf{v}}_i \in \mathbb{R}^k$, and minimize the reconstruction objective (Eq. (4)) by optimizing over $\hat{\mathbf{v}}_i$ instead of $\hat{\mathbf{x}}_i$. Explicitly, we minimize the loss

$$\mathcal{L}_{\text{rec}}\left(\{\hat{\alpha}_{i,c}\}_{i \in [n], c \in [C]} \bigcup \{\Psi\left(\hat{\mathbf{v}}_i\right)\}_{i \in [n]}\right).$$

Perhaps the simplest way to perform this dimensionality reduction is with PCA. As can be seen in Fig. 6, reducing the dimension by a certain factor with PCA allows to reduce the number of query points by a similar factor while maintaining a similar reconstruction quality.

We may also consider reducing the dimension with an autoencoder. Specifically, we let $\Psi$ be the decoder from a pretrained VAE. For our experiments, we choose TAESD3 (Bohan, 2023), a small, distilled version of the VAE used in Stable Diffusion 3 (Esser et al., 2024). The full set of reconstructions can be found in Fig. 11.

## 7 Experiments

**Datasets.** For our experiments, we use the CIFAR10, CIFAR100 (Krizhevsky et al., 2009), and celebA (Liu et al., 2015) datasets. (Results on CIFAR100 are shown in the Appendix.) Following the convention set by previous papers on dataset reconstruction (Haim et al., 2022; Loo et al., 2023; Oz et al., 2024; Buzaglo et al., 2024), we set the number of samples at $N = 500$. To allow for more samples $\mathbf{z}_j$ to be used for loss in Eq. (4), we redistribute the original train-test split. Furthermore, for the CIFAR10 dataset, we obtain more query points $\mathbf{z}_j$ by leveraging synthetic data. Specifically, we use the CIFAR-5M dataset (Nakkiran et al.,

2020) that includes roughly 6 million artificially generated images that are similar to CIFAR10. Importantly, we picked the images used to train $f$ so that they were not in the training data used for the model that generated CIFAR-5M.

**Metrics.** Following Haim et al. (2022); Loo et al. (2023) we use both Structural Dissimilarity (DSSIM) and $L_2$ as our main metric to indicate the quality of our reconstructions. DSSIM is based on SSIM (Wang, 2004) and defined as $\mathrm{DSSIM}(\mathbf{x}, \mathbf{x}') = \frac{1 - \mathrm{SSIM}(\mathbf{x}, \mathbf{x}')}{2}$. In Haim et al. (2022), a DSSIM smaller than 0.3 was chosen as a threshold for a good reconstruction, and we follow this convention as well. We remark that feature-based approaches such as LPIPS (Zhang et al., 2018) may be less meaningful for data reconstruction attacks when the reconstructions fall outside the distribution of natural images. As in prior works, most of our reconstructions look like a noisy version of an image in the training set, and therefore $L_2$ and DSSIM are natural. We report the percentiles computed by finding the nearest reconstruction to each of the training points. To determine the percentage of the dataset reconstructed, we compute the DSSIM between all pairs of reconstructions and training points and count the pairs of mutual nearest neighbors (meaning a reconstruction that is the closest to a certain training point out of all reconstructions and also vice versa).

## 7.1 Comparison to other Reconstruction Schemes

Recent reconstruction works apply to slightly different settings than ours (e.g., Haim et al. (2022); Loo et al. (2023) do not apply to infinite-dimensional features). The following serves as the closest options for comparison, and we highlight some of the key differences between our settings and theirs:

**Loo et al. (2023) RBF.** We adapted the attack of Loo et al. (2023) to kernel regression with the RBF kernel. Specifically, since their attack requires computing the feature map explicitly, it is not directly applicable in infinite-dimensional feature spaces. To overcome this, we apply their attack to a random Fourier feature approximation of the RBF kernel (Rahimi & Recht, 2007) using 400k random features. Their method further initializes twice the number of intended reconstruction candidates ($n = 1000$) since this improved their results. We also note that in this setting, the attack for MSE-loss trained models of Haim et al. (2022); Buzaglo et al. (2024) is similar to the attack of Loo et al. (2023).

**Haim et al. (2022); Buzaglo et al. (2024).** These KKT-based attacks are designed for neural networks and achieve good results. However, in contrast to our method that is query-only, they assume access to the model parameters. We note that implementing their KKT attacks for infinite-dimensional kernels is infeasible. Nevertheless, these do serve as a rough baseline for what metrics one should expect, and we compare our attack to theirs in Appendix D.1. Our reconstruction attack achieves scores on par with or better than their baselines across all metrics.

We report the results in Table 1. The comparison is carried out on models trained on the same images of CIFAR10. In the most similar comparison, our method on RBF kernels outperforms the corresponding comparison with (Loo et al., 2023) by a large margin despite being a query-only method. We also outperform or achieve comparable performance to all other methods. Strikingly, our results indicate that merely hiding parameters is an insufficient defense mechanism for data reconstruction attacks. In Table 2, we further compare different kernels and use cases on the celebA dataset. This includes Laplace and RBF kernels trained by either KRR or SVM. We also list the results when reconstructing using a VAE. In Table 3, we verify that our method is not sensitive to knowing the exact number of training samples $n$. In fact, using a larger number of reconstruction candidates allows us to reconstruct a larger portion of the dataset.

Table 1: Comparison of different methods on CIFAR-10, $n = N = 500$. Our reconstructions here are for kernel regression. The best result in each column is in bold, and the second best is underlined.

| Method | % of Dataset Reconstructed ↑ | | DSSIM ↓ | | | $L_2$ ↓ | | |
|---|---|---|---|---|---|---|---|---|
| | Total | DSSIM < 0.3 | 25% | 50% | 75% | 25% | 50% | 75% |
| (Loo et al., 2023) RBF (Approximation) | 46.2% | 42.8% | 0.209 | 0.311 | 0.369 | 4.484 | 7.978 | 10.738 |
| Ours RBF | 67.8% | 62.2% | 0.12 | 0.232 | 0.351 | 2.145 | 4.091 | 10.746 |
| Ours Laplace | **81.2**% | **77%** | **0.079** | **0.154** | **0.258** | **1.621** | **2.898** | **6.028** |
| Ours NTK | 23.2% | 4.6% | 0.343 | 0.379 | 0.405 | 12.870 | 14.850 | 16.350 |
| Ours Cubic Polynomial | 26.0% | 24.2% | 0.286 | 0.329 | 0.359 | 7.735 | 10.326 | 12.384 |

Table 2: Reconstructions from the celebA dataset with our method, with various kernels and tasks. The best result in each column is in bold, second best is underlined.

| Method | % of Dataset Reconstructed ↑ | | DSSIM ↓ | | | $L_2$ ↓ | | | Resolution |
|---|---|---|---|---|---|---|---|---|---|
| | Total | DSSIM < 0.3 | 25% | 50% | 75% | 25% | 50% | 75% | |
| Laplace KRR | **81.2** % | 57% | 0.239 | 0.289 | 0.328 | 7.770 | **10.356** | **13.848** | $64 \times 64$ |
| RBF KRR | 57.4% | 40.4% | 0.224 | 0.338 | 0.378 | **6.423** | 12.268 | 21.585 | $64 \times 64$ |
| Laplace SVM | 25.0% | 1.8% | 0.369 | 0.39 | 0.405 | 16.900 | 19.523 | 23.439 | $64 \times 64$ |
| RBF SVM | 50.8% | 3.4% | 0.353 | 0.381 | 0.400 | 13.052 | 16.665 | 25.753 | $64 \times 64$ |
| Laplace KRR (using VAE) | 79.6% | **78.6%** | **0.162** | **0.205** | **0.253** | 14.031 | 18.973 | 28.688 | $128 \times 128$ |
| RBF KRR (using VAE) | 60.2% | 58.6% | 0.17 | 0.245 | 0.287 | 14.88 | 23.778 | 41.163 | $128 \times 128$ |

Table 3: Training data recovery when the size of the training data $N = 500$ is unknown. Each row in the table shows the optimization of Eq. (4) with $n \in \{300, 400, 600, 700\}$. The metrics are calculated with respect to the top $\min(n, N)$ reconstructions. The attacked model is trained for kernel regression with the Laplace kernel on CIFAR10.

| Reconstruction Candidates | % of Dataset Reconstructed ↑ | | DSSIM ↓ | | | $L_2$ ↓ | | |
|---|---|---|---|---|---|---|---|---|
| | Total | DSSIM < 0.3 | 25% | 50% | 75% | 25% | 50% | 75% |
| 300 | 83.33% | 67.67% | 0.195 | 0.327 | 0.384 | 3.163 | 7.787 | 12.543 |
| 400 | 82.25% | 71.5% | 0.118 | 0.235 | 0.353 | 2.088 | 4.607 | 10.825 |
| 600 | 92.2% | 91.2% | 0.05 | 0.091 | 0.157 | 1.225 | 1.964 | 3.25 |
| 700 | **96.6%** | **96.6%** | **0.038** | **0.069** | **0.114** | **1.014** | **1.652** | **2.544** |

Table 4: Training data recovery from CIFAR10 when the attacked model is unknown, but can be represented as $\boldsymbol{k}(\mathbf{x}, \mathbf{x}') = \exp\left(-\gamma \|\mathbf{x} - \mathbf{x}'\|_2^\beta\right)$ for some $\beta \geq 1, \gamma > 0$. This family encompasses Laplace kernels (with $\beta = 1$) and RBF kernels (with $\beta = 2$). We randomly initialize $\hat{\beta}$ and $\hat{\gamma}$, and optimize them with the goal of approximating $\beta$ and $\gamma$. In all cases, $\hat{\beta}$ was initialized as 0.15 and $\hat{\gamma}$ at 0.01. Both Laplace kernels and RBF kernels are well reconstructed without exact knowledge of the kernel being attacked.

| Task | Final $\hat{\beta}$ | Target $\beta$ | Final $\hat{\gamma}$ | Target $\gamma$ | % of Dataset Reconstructed ↑ | | DSSIM ↓ | | | $L_2$ ↓ | | |
|---|---|---|---|---|---|---|---|---|---|---|---|---|
| | | | | | Total | DSSIM < 0.3 | 25% | 50% | 75% | 25% | 50% | 75% |
| SVM | 0.982 | 1 | 0.165 | 0.15 | 88.4% | 82.4% | 0.117 | 0.161 | 0.245 | 2.079 | 2.955 | 5.36 |
| KRR | 1.039 | 1 | 0.124 | 0.15 | 86.6% | 82.4% | 0.055 | 0.115 | 0.224 | 1.358 | 2.321 | 5.05 |
| SVM | 1.991 | 2 | 0.0309 | 0.003 | 54% | 32% | 0.27 | 0.335 | 0.4 | 4.61 | 7.894 | 12.691 |
| KRR | 1.991 | 2 | 0.00307 | 0.003 | 58.2% | 48.8% | 0.197 | 0.299 | 0.379 | 3.093 | 6.299 | 12.366 |

## 8 Discussion and Limitations

We presented a data reconstruction attack for kernel methods that works by querying the attacked model at various points.

We proved uniqueness for strictly p.d. kernels and showed that minimizing our reconstruction loss implies reconstructing the training data. An interesting direction for future work is to provide a more complete

picture of the loss landscape, specifically, to characterize the difficulty of reaching *near* a minimizer of the loss.

It is also worthwhile to note that all reconstruction attacks in this paper were performed on a single GPU. Since query points do not require labels, and natural images are abundant, the reconstruction loss could potentially be computed with several orders of magnitude more query points than in this paper. We thus see expanding this attack to the scale found in state of the art models as an interesting avenue for future research.

Lastly, we note that all datasets used in this work are common and public. In particular, no sensitive private data is exposed throughout this work. We believe that the potential benefits of shedding light on potential privacy risks by publicizing this reconstruction attack far outweigh any potential risks.

**Broader Impact Statement**

This paper presents a data reconstruction attack that sheds light on potential privacy risks. All datasets used in this work are common and public. In particular, no sensitive private data is exposed throughout this work. We believe that the potential benefits of publicizing this reconstruction attack far outweigh any potential risks.

## Acknowledgements

Research was partially supported by the Israeli Council for Higher Education (CHE) via the Weizmann Data Science Research Center, by the Knell Family Institute for Artificial Intelligence, by the MBZUAI-WIS Joint Program for Artificial Intelligence Research, and by research grants from the Estates of Tully and Michele Plesser and the Anita James Rosen and Harry Schutzman Foundations.

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

# A    Implementation Details

Below, we detail the implementation details. Every reconstruction attack in this paper was run on a single NVIDIA H100 GPU. Runtimes varied, but as a reference, attacks on CIFAR10 that used $m = 500,000$ and ran for $300,000$ gradient descent steps ran for roughly 11 hours. We did not try to optimize runtimes, and saw in our internal experiments that many fewer query points and gradient steps are necessary for decent results. As such, we infer that good results can also be obtained in a fraction of that time.

## A.1    Hyperparameters of the Reconstruction Algorithm

Unless specifically stated otherwise, all runs in the paper were with the hyper parameters listed below.

**Number of Query Points**    We use $m = 500,000$ for CIFAR10, $m = 200,000$ for CIFAR100 (generated from $50,000$ images with vertical and horizontal flips) and $m = 162,770$ for celebA.

**Initialization**    We initialize our reconstructions $\hat{\mathbf{x}}_i \overset{i.i.d.}{\sim} N(0, 0.3I_d)$ and $\hat{\alpha}_i \overset{i.i.d.}{\sim} N(0, 0.05I_d)$ unless specifically stated otherwise. For the NTK we initialized $\hat{\alpha}_i \overset{i.i.d.}{\sim} N(0, 0.5I_d)$

**Optimization Process**    We use Adam optimizer (Kingma, 2014) with $\beta_1 = 0.9, \beta_2 = 0.99$. We use OneCycle (Smith & Topin, 2019) learning rate scheduler with a maximal base rate of $2e-2$ for the reconstructions and $1e-2$ for $\hat{\alpha}_i$, warmup of 15% of the optimization process, div factor of 10, and final div factor of 100 and three phase scheduling. We run our attack for 300,000 steps.

## A.2    Data Representation

For the CIFAR10 and CIFAR100 (Krizhevsky et al., 2009) datasets we consider the labels as one-hot vectors normalized to have zero mean and unit variance. For the celebA (Liu et al., 2015) resize and crop the images to be at a resolution of $64 \times 64$. When using a VAE for the reconstruction we resize celebA images to $128 \times 128$. The task in this dataset is multi-label classification, and thus, we represent the label of each of the 40 attributes as $\pm 1$.

## A.3    Kernels

The precise formulation of all kernels used in this paper can be found in Appendix C. For the Laplace kernel, we use $\gamma = 0.15$ for $32 \times 32$ images and 0.03 for $64 \times 64$ or $128 \times 128$. For the RBF, we use $\gamma = 0.003$ for $32 \times 32$ images, 0.0005 for $64 \times 64$ and 0.0001 for $128 \times 128$. For the polynomial kernel, we take a cubic polynomial ($\ell = 3$) with $c_0 = 1$ and $\gamma = 0.001$. For the NTK we use $L=3$.

## A.4    Training the Attacked Model

For $f$ trained with kernel regression, we compute the minimizer explicitly as written in Sec. 4.2. For SVM, as briefly mentioned in Sec. 4.2, we express $f$ as in Eq. (2) and optimize $\alpha_{i,c}$ on the hinge loss by gradient descent. We perform $100,000$ iterations with a learning rate of $1e-2$ and OneCycle scheduler. This ensures that in all of the experiments in this paper, the parameters converge.

## A.5    PCA

If we know the underlying distribution of the data, we can compute a matrix $U \in \mathbb{R}^{d \times k}$ whose $k$ columns are orthonormal vectors corresponding to the most influential directions and define $\Psi(\hat{\mathbf{v}}_i) = U\hat{\mathbf{v}}_i$. For standard image datasets, one may take $k$ to be much smaller than $d$ such that the projections $UU^\top \mathbf{x}$ are visually very similar to the original images.

Nevertheless, when optimizing $\hat{\mathbf{v}}_i$ this way, an undesired side-effect is that the "implicit-bias" of the optimizer is changed. All of our experiments use Adam, which adjusts the learning rate of each parameter being

optimized. In our case, this amounts to adjusting the learning rate per pixel. PCA changes the basis, so now the learning rate is adjusted per principal direction. We found this to be undesired, so instead of initializing $\hat{\mathbf{v}}_i$ to be of dimension $k$, we initialize $\hat{\mathbf{v}}_i$ to be of dimension $d$ and set $\hat{\mathbf{x}}_i = UU^\top \hat{\mathbf{v}}_i$ to be its projection.

### A.6   Special Considerations.

For the polynomial kernel, we add a small loss term that discourages noisy reconstructions. We observe that this is not strictly necessary, but slightly improves results. We also used a learning rate of $5e-3$ for both the reconstructions as well as $\hat{\alpha}_i$. For the NTK we used a base learning rate of $1e-4$ for the reconstructions and $1e-3$ and $\hat{\alpha}$ respectively, and ran the optimization for 1,000,000 steps.

## B   Proofs

### B.1   Uniqueness of Representations

**Proposition 3.** *Let $\boldsymbol{k}$ be a strictly p.d. kernel, and suppose that we have two functions given by*

$$f(\mathbf{z}) = \sum_{i=1}^{N} \alpha_i \boldsymbol{k}(\mathbf{z}, \mathbf{x}_i), \qquad \hat{f}(\mathbf{z}) = \sum_{i=1}^{n} \hat{\alpha}_i \boldsymbol{k}(\mathbf{z}, \hat{\mathbf{x}}_i)$$

*where $\forall i \in [n], i' \in [N], \alpha_i, \hat{\alpha}_{i'} \neq 0$, and $\forall i \neq j \in [n], i' \neq j' \in [N] : \mathbf{x}_i \neq \mathbf{x}_j$ and $\hat{\mathbf{x}}_{i'} \neq \hat{\mathbf{x}}_{j'}$. Then if the functions are equal, i.e, $\forall \mathbf{z} \in \mathcal{X}, f(\mathbf{z}) = \hat{f}(\mathbf{z})$, then $n = N$ and, up to permutation, $\mathbf{x}_i = \hat{\mathbf{x}}_i$ and $\alpha_i = \hat{\alpha}_i$.*

*Proof.* Define $U = \{\mathbf{u}_1, ..., \mathbf{u}_r\} := X \cup \hat{X}$ where $X = \{\mathbf{x}_1, ..., \mathbf{x}_N\}$, $\hat{X} = \{\hat{\mathbf{x}}_1, ...\hat{\mathbf{x}}_n\}$ and $r := \left| X \cup \hat{X} \right| \leq n+N$.

Rewrite the difference $f(\mathbf{z}) - \hat{f}(\mathbf{z}) = \sum_{i=1}^{N} \alpha_i \boldsymbol{k}(\mathbf{z}, \mathbf{x}_i) - \sum_{j=1}^{n} \hat{\alpha}_j \boldsymbol{k}(\mathbf{z}, \hat{\mathbf{x}}_j)$ as $\sum_{i=1}^{r} b_i \boldsymbol{k}(\mathbf{z}, \mathbf{u}_i)$ for suitable $b_i \in \mathbb{R}$. Under our assumption that $f \equiv \hat{f}$, for $j = 1, ..., r$, substituting $\mathbf{z} = \mathbf{u}_j$ we get $\sum_{i=1}^{r} b_i \boldsymbol{k}(\mathbf{u}_j, \mathbf{u}_i) = 0$. That is, we have the linear system $K_U \boldsymbol{b} = 0$, where $K_U$ is the kernel matrix of $U$.

Since $K_U$ is strictly positive definite, it is invertible, so we conclude $\boldsymbol{b} = 0$. Notice each $b_i$ is either $\alpha_i$ for some $i \in [n]$, $-\hat{\alpha}_j$ for some $j \in [N]$, or $\alpha_i - \hat{\alpha}_j$ for some $i, j$. Because of the assumption that $\alpha_i, \hat{\alpha}_j$ are nonzero, only the latter case is possible. This means each $\mathbf{x}_i$ and $\hat{\mathbf{x}}_j$ belong to the intersection $X \cap \hat{X}$, so $X \subseteq X \cap \hat{X}, \hat{X} \subseteq X \cap \hat{X}$, implying $X = X \cap \hat{X} = \hat{X} = U$. So, $r = n = N$ and, assuming w.l.o.g that $X$ and $\hat{X}$ are indexed in the same order, $b_i = \alpha_i - \hat{\alpha}_i$. Since $b_i = 0$ this gives $\alpha_i = \hat{\alpha}_i$, so we are finished.

$\square$

### B.2   Proof of Thm. 2

**Definition 4.** A function $f : \mathbb{R}^d \to \mathbb{R}$ is said to be *analytic* on an open set $U$ if for every $\mathbf{x} \in U$, $f$ is given as a convergent power series in a neighborhood $V \subseteq U$ of $\mathbf{x}$. We say $f$ is analytic at $\mathbf{x}$ if $f$ is analytic in some neighborhood of $\mathbf{x}$.

**Definition 5.** Let $\mathcal{X}$ be any input space, and $\boldsymbol{k} : \mathcal{X} \times \mathcal{X} \to \mathbb{R}$ a kernel. The *evaluation function* of $\boldsymbol{k}$ is the function $g(X, \boldsymbol{\alpha}, \mathbf{z}) = \sum_{i=1}^{n} \alpha_i \boldsymbol{k}(\mathbf{x}_i, \mathbf{z})$ for a training set $X = (\mathbf{x}_1, ..., \mathbf{x}_n) \in \mathcal{X}^n$, coefficients $\boldsymbol{\alpha} \in \mathbb{R}^n$, and a point $z \in \mathcal{X}$. We denote also $g_{X,\boldsymbol{\alpha}}(z) = g(X, \boldsymbol{\alpha}, \mathbf{z})$ and write $g_{X,\boldsymbol{\alpha}} \not\equiv 0$ if $\exists z \in \mathcal{X} : g_{X,\boldsymbol{\alpha}}(z) \neq 0$.

Recall Def. (1) of an *almost analytic* kernel.

We restate Thm. 2 to include explicit guarantees on the values of the coefficients $\hat{\alpha}_{i,c}$. Notice this statement implies Thm. 2.

**Theorem 6.** *Let $\mathcal{X} \subseteq \mathbb{R}^d$ be open, and $\boldsymbol{k} : \mathcal{X} \times \mathcal{X} \to \mathbb{R}$ be an almost analytic, strictly p.d. kernel. Let $\mathcal{D}$ be any distribution given by a density over $\mathcal{X}$. Let $f_c = \sum_{i=1}^{N} \alpha_{i,c} \boldsymbol{k}(\mathbf{z}, \mathbf{x}_i)$ for $c \in [C]$ be $C$ predictors as in Eq. (2), where $\forall i \neq j \in [N] : \mathbf{x}_i \neq \mathbf{x}_j$, and $\forall i \in [N], \exists c \in [C] : \alpha_{i,c} \neq 0$. Let $n \geq N$ and $m > n(d+2)$, and let $\hat{\alpha}_{i,c}, \hat{\mathbf{x}}_i$ be any solution to the minimization problem defined by the reconstruction loss in Eq. (4). Then it*

*holds with probability 1 over $Z = (\mathbf{z}_1, ..., \mathbf{z}_m) \sim \mathcal{D}^m$, up to permutation and summation of terms of identical $\hat{\mathbf{x}}_i$, that $\forall i \in [N], c \in [C] : \hat{\alpha}_{i,c} = \alpha_{i,c}, \hat{\mathbf{x}}_i = \mathbf{x}_i$, and $\forall N < i \le n, c \in [C] : \hat{\alpha}_{i,c} = 0$.*

*Remark* 7. The theorem is stated with simple yet commonly satisfied conditions, namely a strictly p.d., analytic kernel on $\mathbb{R}^d$, allowing isolated non-analytic points. Nonetheless, we provide some generalizations.

1. **Generalization to submanifolds**: In the statement of Thm. 6, we require $\mathcal{X}$ to be an open subset of $\mathbb{R}^d$. However, the main part of the proof, in which we show that the set of $m$-tuples $Z = (z_1, ..., z_m)$ for which uniqueness does not hold is contained in a submanifold of $\mathcal{X}^m$ of dimension $< md$, relies only on the fact that $\mathcal{X}$ is a smooth manifold (for the Submersion Level Set Theorem). To further conclude that the probability of sampling such a $Z$ is 0, we require only a well-defined distribution $\mathcal{D}$ on $\mathcal{X}$ which gives 0 probability to submanifolds of dimension strictly smaller than $d$. Therefore, the same proof strategy would work for any submanifold $\mathcal{X} \subseteq \mathbb{R}^{d'}$ of dimension $d$ with such a properly defined distribution (for instance, a Riemannian manifold with $\mathcal{D}$ defined by a density, integrated with regard to the canonical volume measure). A common example would be distributions over the sphere $\mathbb{S}^d$. Note that in general, analytic functions are only properly defined on real-analytic manifolds; see the next point for a generalization to $C^\infty$ kernels.

2. **Generalization to $C^\infty$ kernels**: The requirement on $\boldsymbol{k}$ may be weakened so that $\boldsymbol{k} \in C^\infty(\mathcal{X})$, under the condition that any linear combination $f(z) = \sum_{i=1}^n \alpha_i \boldsymbol{k}(z, x_i)$ which is not identically zero, does not have partial derivatives of every order vanishing at a point $z$ where $f(z) = 0$ (this is a key property of analytic functions which is useful in the proof of Thm. 2). In this case, if every such $f(\mathbf{z})$ has isolated points $\mathbf{z}$ where $f$ is not $C^\infty$ or has vanishing derivatives of every order, the number of queries required is $m > n(d+1) + \frac{n(d+1)}{d}$.

3. **Generalization to larger sets of non-analytic points**: One may have a kernel $\boldsymbol{k}$ so that for every fixed $\mathbf{x}$, $\boldsymbol{k}(\mathbf{x}, \mathbf{z})$ has non-analytic points $\mathbf{z}$ in a submanifold $\mathcal{M}_\mathbf{x} \subseteq \mathcal{X}$ of dimension $d' < d$. This too can be accommodated in the theorem, as long as these non-analytic points are smoothly parameterized by $\mathbf{x}$. The number of queries in this case would be $m > n(d+1) + \frac{nd}{d-d'}$. In the $C^\infty$ case, in which we consider for every linear combination $f(\mathbf{z})$, points $\mathbf{z}$ from a $d'$-dimensional manifold where $f(\mathbf{z})$ has all partial derivatives vanishing, we would require $m > n(d+1) + \frac{n(d+1)}{d-d'}$. Notice the bounds for the isolated case are recovered with $d' = 0$. For an empty set (a kernel that is analytic everywhere) the proof holds with $m > n(d+1)$.

### B.2.1 Proof of Thm. 6

*Proof.* First, we assume w.l.o.g $C = 1$. Indeed, any solution attaining 0 loss for Eq. (4) also attains 0 loss for each $c \in C$. Therefore, if the theorem holds for $C = 1$ we can apply it separately for every $c \in C$ and obtain that for $m > n(d+2)$, it holds with probability 1 over $Z \sim \mathcal{D}^m$ that the coefficients $\hat{\alpha}_{i,c}$ and training points $\hat{\mathbf{x}}_i$ reflect the true training data (note that the intersection of all $C$ events still holds w.p. 1). Since $\forall i \in [N]$ there exists $c \in [C]$ with $\alpha_{i,c} \ne 0$, every $\mathbf{x}_i$ will appear in the solution set for the appropriate $c$, but since the $\hat{\mathbf{x}}_i$ are shared between reconstruction terms, it will appear in all of them. Hence $\{\mathbf{x}_1, ..., \mathbf{x}_N\} \subseteq \{\hat{\mathbf{x}}_1, ..., \hat{\mathbf{x}}_n\}$. W.l.o.g we permute $1, ..., n$ so that $\forall 1 \le i \le N : \hat{\mathbf{x}}_i = \mathbf{x}_i$, and assume that $\hat{\mathbf{x}}_1, ..., \hat{\mathbf{x}}_n$ are distinct (if they are not, for each $c \in [C]$ we sum terms for identical $\hat{\mathbf{x}}_i$, and complete the set $\{\hat{\mathbf{x}}_1, ..., \hat{\mathbf{x}}_n\}$ to $n$ distinct vectors with a choice of arbitrary $\hat{\mathbf{x}}_i$ and coefficients equal to 0). Now, observing each $c \in [C]$ separately, the equality of the coefficients $\forall 1 \le i \le N : \hat{\alpha}_{i,c} = \alpha_{i,c}$ and $\forall i > N : \hat{\alpha}_{i,c} = 0$ follows from the case $C = 1$.

Assuming $C = 1$, for ease of notation we denote $f = f_1$ and $\forall i \in [N] : \alpha_i = \alpha_{i,1}$. Since $\boldsymbol{k}$ is strictly p.d., from Proposition 3, it suffices to show that for any minimizer $\hat{f} = \sum_{i=1}^n \hat{\alpha}_i \boldsymbol{k}(\cdot, \hat{\mathbf{x}}_i)$ with $\forall j \in [m] : \hat{f}(z_j) = f(z_j)$ it holds that $f \equiv \hat{f}$. Denote $V$ the family of training sets which define predictors non-equivalent to $f$:

$$V = \{(\hat{X}, \hat{\boldsymbol{\alpha}}) \in \mathcal{X}^n \times \mathbb{R}^n \mid \exists \mathbf{z} \in \mathcal{X} : \sum_{i=1}^n \hat{\alpha}_i \boldsymbol{k}(\mathbf{z}, \hat{\mathbf{x}}_i) \ne f(\mathbf{z})\}$$

$V$ is clearly an open set since, if for some $(\hat{X}, \hat{\boldsymbol{\alpha}})$ there exists $\mathbf{z} \in \mathcal{X}$ with $\sum_{i=1}^n \hat{\alpha}_i \boldsymbol{k}(\mathbf{z}, \hat{\mathbf{x}}_i) \ne f(\mathbf{z})$, the same holds for a neighborhood of $(\hat{X}, \hat{\boldsymbol{\alpha}})$ from continuity of $\boldsymbol{k}$.

Note that any $\hat{f}$ of the above form with $f \not\equiv \hat{f}$ is represented by some $(\hat{X}, \hat{\boldsymbol{\alpha}}) \in V$.

Define $g_f : V \times \mathcal{X} \to \mathbb{R}$ as

$$g_f\left((\hat{X}, \hat{\boldsymbol{\alpha}}), \mathbf{z}\right) = \sum_{i=1}^{n} \hat{\alpha}_i \boldsymbol{k}(\mathbf{z}, \hat{\mathbf{x}}_i) - f(\mathbf{z}) = \sum_{i=1}^{n} \hat{\alpha}_i \boldsymbol{k}(\mathbf{z}, \hat{\mathbf{x}}_i) - \sum_{i=1}^{N} \alpha_i \boldsymbol{k}(\mathbf{z}, \mathbf{x}_i) \tag{8}$$

Notice that $g_f((\hat{X}, \hat{\boldsymbol{\alpha}}), \mathbf{z})$ is in fact the evaluation function on a training set of size $\leq 2n$ defined by $(\hat{X} * X, \hat{\boldsymbol{\alpha}} * (-\boldsymbol{\alpha}))$, where $*$ signifies concatenation. Also, for any $(\hat{X}, \hat{\boldsymbol{\alpha}})$, $g_f((\hat{X}, \hat{\boldsymbol{\alpha}}), \cdot) \not\equiv 0$ by definition of $V$.

From here on we denote $\boldsymbol{v} = (\hat{X}, \hat{\boldsymbol{\alpha}}) \in V \subseteq \mathbb{R}^{n(d+1)}$.

Denote $G : V \times \mathcal{X}^m \to \mathbb{R}^m$ the function $g_f$ evaluated on $m$ points:

$$G(\boldsymbol{v}, Z) = (g_f(\boldsymbol{v}, \mathbf{z}_1), ..., g_f(\boldsymbol{v}, \mathbf{z}_m))$$

And $\pi : V \times \mathcal{X}^m$ the projection onto $Z$:

$$\pi(\boldsymbol{v}, Z) = Z$$

Denote

$$\mathcal{Z} = \{Z \in \mathcal{X}^m \mid \exists \boldsymbol{v} \in V : G(\boldsymbol{v}, Z) = 0\}$$
$$= \pi(G^{-1}(0))$$

The rest of the proof will be dedicated to showing that when $m$ is large enough, the set $\mathcal{Z}$ is a Lebesgue null set in $\mathcal{X}^m$. Indeed, with this result, since $\mathcal{D}$ is a distribution with density, it follows that $\mathcal{Z}$ has an integrated density of 0. It follows that with probability 1 over $Z \sim \mathcal{D}^m$, there exists no suitable $\hat{f} \not\equiv f$ with $\forall j \in [m] : \hat{f}(\mathbf{z}_j) = f(\mathbf{z}_j)$, thereby finishing the proof.

First we show we can discount the non-analytic points. We say $(\boldsymbol{v}, \mathbf{z})$ is "violating" if $g_f(\boldsymbol{v}, \cdot)$ is not analytic at $\mathbf{z}$. We prove that the set of $m$-tuples of queries $Z \in \mathcal{X}^m$, so that there exists a $\boldsymbol{v} \in V$ for which more than $n$ of the queries are violating, is null. This will allow us to assume that w.p. 1, $Z$ contains more than $n(d+1)$ non-violating queries for every possible $\boldsymbol{v} \in V$. Denote

$$\mathcal{Z}_B = \{Z \in \mathcal{X}^m \mid \exists \boldsymbol{v} \in V, I \subseteq [m], |I| = n' > n : \forall j \in I, (\boldsymbol{v}, \mathbf{z}_j) \text{ is violating }\}$$

We show $\mathcal{Z}_B$ is null.

Since $\boldsymbol{k}$ is almost analytic, there exist suitable functions $\{\Gamma_s : \mathcal{X} \to \mathcal{X}\}_{s \in \mathbb{N}}$ so that $\boldsymbol{k}(\mathbf{x}, \cdot)$ is analytic for every $\mathbf{z} \notin \bigcup_{s=1}^{\infty}\{\Gamma_s(\mathbf{x})\}$. Since analyticity is preserved under linear combinations, $g_f((\hat{X}, \hat{\boldsymbol{\alpha}}), \cdot)$ is analytic at every $\mathbf{z} \notin \{\Gamma_s(\mathbf{x}) \mid s \in \mathbb{N}, \mathbf{x} \in \{\mathbf{x}_1, ..., \mathbf{x}_N, \hat{\mathbf{x}}_1, ..., \hat{\mathbf{x}}_n\}\}$.

For any $I \subseteq [m], |I| = n' > n$, denote $\mathcal{Z}_{B,I}$ the set of $Z \in \mathcal{X}^m$ where $\exists \boldsymbol{v} \in V : \forall j \in I : (\boldsymbol{v}, \mathbf{z}_j)$ is violating. Since $\mathcal{Z}_{\mathcal{B}} = \bigcup_{I \subseteq [m], |I| > n} \mathcal{Z}_{B,I}$, it suffices to fix $I$ and show that $\mathcal{Z}_{B,I}$ is null.

For every $(\boldsymbol{v}, Z)$ with violating indices at $I$, it holds that each query $\mathbf{z}_j, j \in I$ is in the image of some $\Gamma_s$ for some $\mathbf{x} \in \{\mathbf{x}_1, ..., \mathbf{x}_N, \hat{\mathbf{x}}_1, ..., \hat{\mathbf{x}}_n\}$. Therefore, the $n'$-tuple $(z_j)_{j \in I}$ is in the image of the following function for some $S = (s_1, ..., s_{n'}) \in \mathbb{N}^{n'}, J = (j_1, ..., j_{n'}) \in [2n]^{n'}$:

$$\Gamma_{S,J}(\tilde{\mathbf{x}}_1, ..., \tilde{\mathbf{x}}_n, ..., \tilde{\mathbf{x}}_{2n}) = (\Gamma_{s_1}(\tilde{\mathbf{x}}_{j_1}), ..., \Gamma_{s_{n'}}(\tilde{\mathbf{x}}_{j_{n'}})) : \mathcal{X}^{2n} \to \mathcal{X}^{n'}$$

However, notice that in our case $\mathbf{x}_1, ..., \mathbf{x}_N$ are fixed, while only $\hat{\mathbf{x}}_1, ..., \hat{\mathbf{x}}_n$ vary. Therefore it holds that (up to permutation of indices)

$$\mathcal{Z}_{B,I} \subseteq \bigcup_{S \in \mathbb{N}^{n'}, J \in [2n]^{n'}} \mathcal{X}^{m-n'} \times \Gamma_{S,J}(\mathcal{X}^n \times \{\hat{\mathbf{x}}_1, ..., \hat{\mathbf{x}}_n\})$$

Fixing $J$ and $S$, it suffices to show that $\Gamma_{S,J}(\mathcal{X}^n \times \{\hat{\mathbf{x}}_1, ..., \hat{\mathbf{x}}_n\})$ is a null set in $\mathcal{X}^{n'}$. This holds simply because $\Gamma_{S,J}$ with $n$ fixed input coordinates is a $C^1$ function from $\mathcal{X}^n$, which is a smooth manifold of dimension $nd$, to a smooth manifold of larger dimension $n'd$.

From here on, for simplicity of notation we assume w.l.o.g that every $(\boldsymbol{v}, Z)$ for $\boldsymbol{v} \in V, Z \in \mathcal{Z}$ contains only non-violating queries; this is justified since we will only need the fact $m > n(d+1)$, and we have shown that, up to a null set of tuples $Z \in \mathcal{X}^m$, each $Z$ has, for any $\boldsymbol{v} \in V$, more than $n(d+1)$ queries which are non violating. The reduction to the non-violating case can be formally achieved through introducing into the parameter $\boldsymbol{r}$ (see below) also the indices $j$ for which $(\boldsymbol{v}, \mathbf{z}_j)$ are violating. Then, around every $(\boldsymbol{v}, Z)$ we select the component functions $g_f(\boldsymbol{v}, \mathbf{z}_j)$, corresponding to non-violating $\mathbf{z}_j$. As we shall see later in the proof, this selection can only increase the size of the solution set, hence if the increased solution set is null, so is the original.

It suffices to show that for every point $(\boldsymbol{v}, Z) \in G^{-1}(0)$, there exists a neighborhood $A$ of $(\boldsymbol{v}, Z)$ for which $\pi(A \cap G^{-1}(0))$ is a null set in $\mathcal{X}^m$. Indeed, taking such neighborhoods for every $(\boldsymbol{v}, Z) \in G^{-1}(0)$, we can take a countable covering subset of these neighborhoods, from second countability of $V \times \mathcal{X}^m$. We obtain that $\pi(G^{-1}(0))$ is the countable union of null sets, hence a null set.

To show this we use the Submersion Level Set Theorem (see for example Lee (2012)).

We first separate $\mathcal{Z}$ into subsets by vanishing of derivatives of the $\mathbf{z}_j$. For $\boldsymbol{v} \in V, \mathbf{z} \in \mathcal{X}$ with $g_f(\boldsymbol{v}, \mathbf{z}) = 0$, denote $r(\boldsymbol{v}, \mathbf{z})$ the integer $r$ for which all partial derivatives of order $\leq r$ of $g_f(\boldsymbol{v}, \cdot)$ vanish at $\mathbf{z}$, and at least one partial derivative of order $r+1$ does not vanish. It holds that $r$ is well-defined since $g_f(\boldsymbol{v}, \cdot)$ is analytic at $\mathbf{z}$ and $g_f(\boldsymbol{v}, \cdot) \not\equiv 0$. For every $\boldsymbol{r} = (r_1, ..., r_m) \in (\mathbb{N} \cup \{0\})^m$ denote

$$\mathcal{Z}_{\boldsymbol{r}} = \{Z \in \mathcal{X}^m \mid \exists \boldsymbol{v} \in V : G(\boldsymbol{v}, Z) = 0 \text{ and } \forall j \in [m] : r(\boldsymbol{v}, \mathbf{z}_j) = r_j\}$$

It holds that

$$\mathcal{Z} = \bigcup_{\boldsymbol{r} \in (\mathbb{N} \cup \{0\})^m} \mathcal{Z}_{\boldsymbol{r}}$$

And this is a countable union.

Fix $\boldsymbol{r}$. It suffices to show that $\mathcal{Z}_{\boldsymbol{r}}$ is null in $\mathcal{X}^m$. Denote by $G_{\boldsymbol{r}} : V \times \mathcal{X}^m \to \mathbb{R}^{m+d(\boldsymbol{r})}$ the "augmented" function obtained by appending to $G$ the evaluation of the $r_j$-order partial derivatives for every $j$ (concretely, $d(\boldsymbol{r}) = \sum_{j=1}^m d^{r_j}$).

It holds that

$$\mathcal{Z}_{\boldsymbol{r}} \subseteq \pi(G_{\boldsymbol{r}}^{-1}(0))$$

So, we must show for $(\boldsymbol{v}, Z) \in G_{\boldsymbol{r}}^{-1}(0)$ that there exists a neighborhood $A$ of $(\boldsymbol{v}, Z)$ so that $\pi(A \cap G_{\boldsymbol{r}}^{-1}(0))$ is null in $\mathcal{X}^m$.

We claim it suffices to find a $m \times m$ submatrix of $\partial G_{\boldsymbol{r}}/\partial Z$ which has rank $m$. Indeed, if this holds we can define a function $\tilde{G}_{\boldsymbol{r}}(\boldsymbol{v}, Z) : V \times \mathcal{X}^m \to \mathbb{R}^m$ by selecting the component functions of $G_{\boldsymbol{r}}$ corresponding to the rows of the submatrix. Note this amounts to reducing the constraints on $(\boldsymbol{v}, Z)$, namely

$$G_{\boldsymbol{r}}^{-1}(0) \subseteq \tilde{G}_{\boldsymbol{r}}^{-1}(0)$$

Now $\tilde{G}_{\boldsymbol{r}}$ satisfies the conditions of the Submersion Level Set Theorem, implying that there exists a neighborhood $A$ of $(\boldsymbol{v}, Z)$ so that $A \cap \tilde{G}_{\boldsymbol{r}}^{-1}(0)$ is a $n(d+1) + md - m < md$ dimensional manifold, hence $\pi(A \cap \tilde{G}_{\boldsymbol{r}}^{-1}(0))$ is null in $\mathcal{X}^m$, implying the same for $\pi(A \cap G_{\boldsymbol{r}}^{-1}(0))$. Finally, the choice of the $m \times m$ submatrix of rank $m$ arises simply from the fact that $\partial G_{\boldsymbol{r}}/\partial Z$ has block matrix structure ($\partial G_{\boldsymbol{r}}/\partial \mathbf{z}_j \neq 0$ only at row indices related to $\mathbf{z}_j$), and for any $j \in [m]$, there exists at least one $r_j$-order partial derivative, hence one row index, which has a non-zero gradient with regard to $\mathbf{z}_j$, from the assumption $r(\boldsymbol{v}, \mathbf{z}_j) = r_j$ $\qquad \square$

## C Relevant Kernels

The following is a list of kernels that are relevant to this paper:

1. Laplace kernel: $\boldsymbol{k}(\mathbf{x}, \mathbf{x}') := \exp(-\gamma \|\mathbf{x} - \mathbf{x}'\|_2)$ for some $\gamma > 0$.

2. Gaussian (RBF) kernel: $\boldsymbol{k}(\mathbf{x}, \mathbf{x}') := \exp(-\gamma \|\mathbf{x} - \mathbf{x}'\|_2^2)$ for some $\gamma > 0$.

3. Polynomial kernel: $\boldsymbol{k}\left(\mathbf{x}, \mathbf{x}'\right) := \left(c_0 + \gamma \left\langle \mathbf{x}, \mathbf{x}' \right\rangle\right)^{\ell}$ for some $c_0, \gamma > 0$ and $\ell \in \mathbb{N}$.

4. Neural Tangent Kernel (NTK): when referring to the NTK, it will always be with respect to a fully connected ReLU network, for which there is a known analytic formula (Jacot et al., 2018; Lee et al., 2019; Bietti & Bach, 2020). First, let

$$\kappa_0(u) := \frac{1}{\pi}(\pi - \arccos(u)), \qquad \kappa_1(u) := \frac{1}{\pi}\left(u\left(\pi - \arccos(u)\right) + \sqrt{1 - u^2}\right).$$

To define the NTK, we first define the $L$ layer Gaussian Process Kernel (GPK; also known as NNGP) on $\mathbb{S}^{d-1}$ as

$$\boldsymbol{k}_{\mathrm{GPK}}^{(L)}(\mathbf{x}, \mathbf{x}') := \kappa_1\left(\boldsymbol{k}_{\mathrm{GPK}}^{(L-1)}(\mathbf{x}, \mathbf{x}')\right), \qquad \boldsymbol{k}_{\mathrm{GPK}}^{(0)}(\mathbf{x}, \mathbf{x}') := \mathbf{x}^\top \mathbf{x}',$$

so that the $L$ layer NTK on $\mathbb{S}^{d-1}$ is

$$\boldsymbol{k}_{\mathrm{NTK}}^{(L)}(\mathbf{x}, \mathbf{x}') := \boldsymbol{k}_{\mathrm{NTK}}^{(L-1)}(\mathbf{x}, \mathbf{x}')\kappa_0\left(\boldsymbol{k}_{\mathrm{GPK}}^{(L-1)}(\mathbf{x}, \mathbf{x}')\right) + \boldsymbol{k}_{\mathrm{GPK}}^{(L)}(\mathbf{x}, \mathbf{x}'), \qquad \boldsymbol{k}_{\mathrm{NTK}}^{(0)} := \boldsymbol{k}_{\mathrm{GPK}}^{(0)}(\mathbf{x}, \mathbf{x}').$$

Now for a general $\mathbf{x}, \mathbf{x}' \in \mathbb{R}^d$, using the fact that for a ReLU activation, the kernel is homogeneous (Bietti & Mairal, 2019), the NTK is given by

$$\boldsymbol{k}_{\mathrm{NTK}}^{(L)}(\mathbf{x}, \mathbf{x}') := \|\mathbf{x}\| \|\mathbf{x}'\| \, \boldsymbol{k}_{\mathrm{NTK}}\left(\frac{\mathbf{x}}{\|\mathbf{x}\|}, \frac{\mathbf{x}'}{\|\mathbf{x}'\|}\right).$$

# D   Further Results

## D.1   Relation to Past Works

The following may serve as additional baselines for recent methods that work in a different setting of neural networks.

**(Haim et al., 2022) / (Buzaglo et al., 2024) NN** - An attack proposed by Haim et al. (2022) and later extended by Buzaglo et al. (2024) on neural networks trained with cross-entropy loss in a way that ensures convergence to KKT points. Their attack utilizes the network weights. It further initializes twice the number of intended reconstruction candidates ($n = 1000$) since this improved their results.

**(Loo et al., 2023) NN** - An extension by Loo et al. (2023) of the KKT attack to neural networks trained near the lazy regime. Their attack uses the weights of the network both at initialization and at the end of training. We compare our results to their attack on two different neural networks, one with a width $W = 1024$ and the other with a width $W = 4096$. This method used $n = 1000$ reconstruction candidates as well.

Table 5: Comparison of different methods on CIFAR-10, $N = 500$. Our reconstructions here are for kernel regression. The best result in each column is in bold, and the second best is underlined.

| Method | % of Dataset Reconstructed ↑ | | DSSIM ↓ | | | $L_2$ ↓ | | | Black-Box |
|---|---|---|---|---|---|---|---|---|---|
| | Total | DSSIM < 0.3 | 25% | 50% | 75% | 25% | 50% | 75% | |
| (Haim et al., 2022) NN | 2.2% | 2.0% | 0.294 | 0.334 | 0.363 | 9.707 | 11.890 | 13.816 | ✘ |
| (Loo et al., 2023) NN-W=1024 | 23.2% | 0.4% | 0.399 | 0.427 | 0.446 | 15.371 | 16.514 | 17.649 | ✘ |
| (Loo et al., 2023) NN-W=4096 | 95.2% | 94.4% | 0.116 | 0.162 | 0.203 | 4.027 | 4.664 | 5.429 | ✘ |
| (Loo et al., 2023) RBF | 46.2% | 42.8% | 0.209 | 0.311 | 0.369 | 4.484 | 7.978 | 10.738 | ✘ |
| Ours RBF | 67.8% | 62.2% | 0.12 | 0.232 | 0.351 | 2.145 | 4.091 | 10.746 | ✔ |
| Ours Laplace | 81.2% | 77% | 0.079 | 0.154 | 0.258 | 1.621 | 2.898 | 6.028 | ✔ |
| Ours NTK | 23.2% | 4.6% | 0.343 | 0.379 | 0.405 | 12.870 | 14.850 | 16.350 | ✔ |
| Ours Cubic Polynomial | 26.0% | 24.2% | 0.286 | 0.329 | 0.359 | 7.735 | 10.326 | 12.384 | ✔ |
| Ours Laplace - $n = 700$ | **96.6%** | **96.6%** | **0.038** | **0.069** | **0.114** | **1.014** | **1.652** | **2.544** | ✔ |

## D.2 Plots

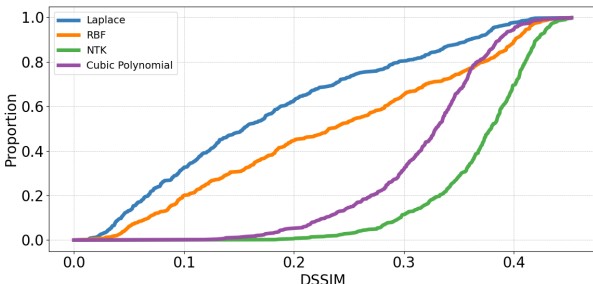

Figure 5: Commulative graph comparing the quality of training data reconstructions for different kernels. The $x$-axis denotes DSSIM, and $y$-axis the proportion of reconstructions whose DSSIM is at most that of the $x$-axis.

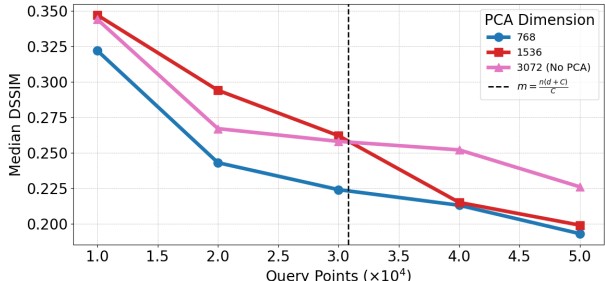

Figure 6: Reconstruction quality with dimensionality reduction with PCA. Models are trained on CIFAR10. Each point represents the quality, measured in median DSSIM, obtained in one run with a different number of query points. Unlike in the rest of the paper, in this experiment, we use $n = 100$ training images.

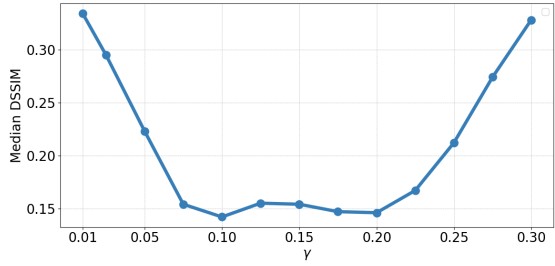

Figure 7: Reconstruction quality, measured in median DSSIM, with the Laplace kernel on CIFAR10 for different choices of $\gamma$.

# E  Full Reconstruction Results

## E.1  Kernel Regression

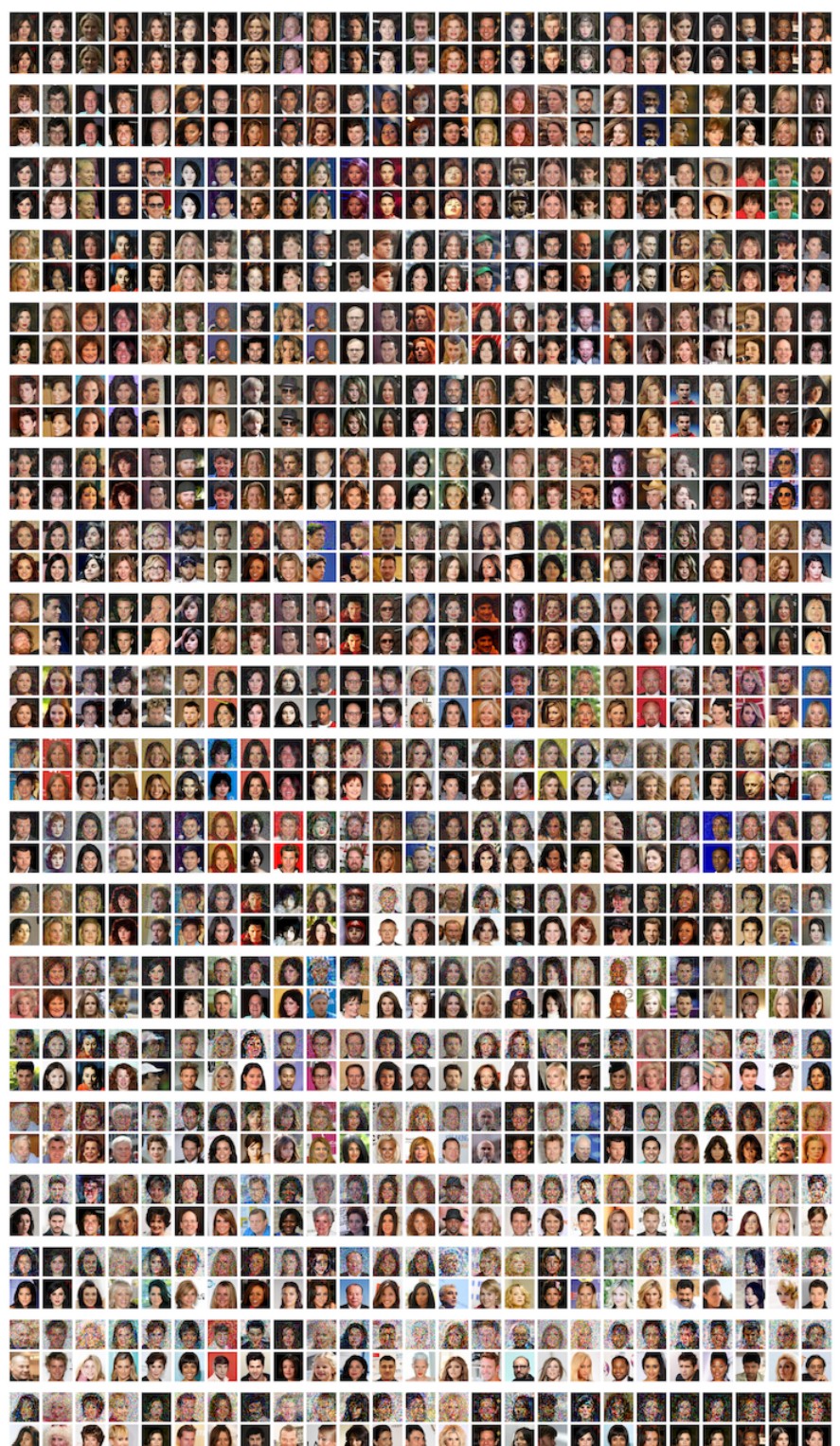

Figure 8: Reconstructions from an RBF kernel trained using kernel regression on celebA.

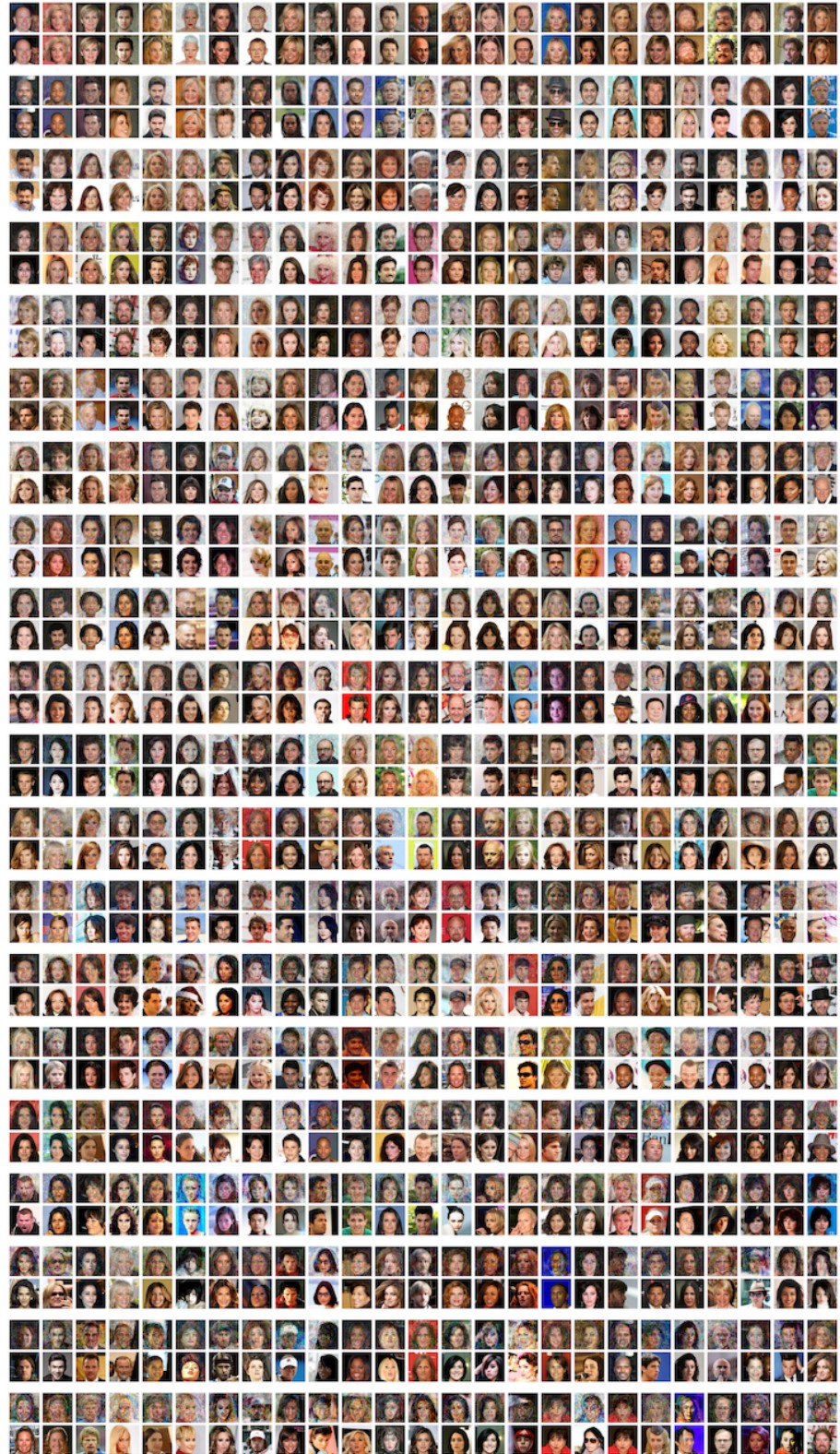

Figure 9: Reconstructions from a Laplace kernel trained using kernel regression on celebA.

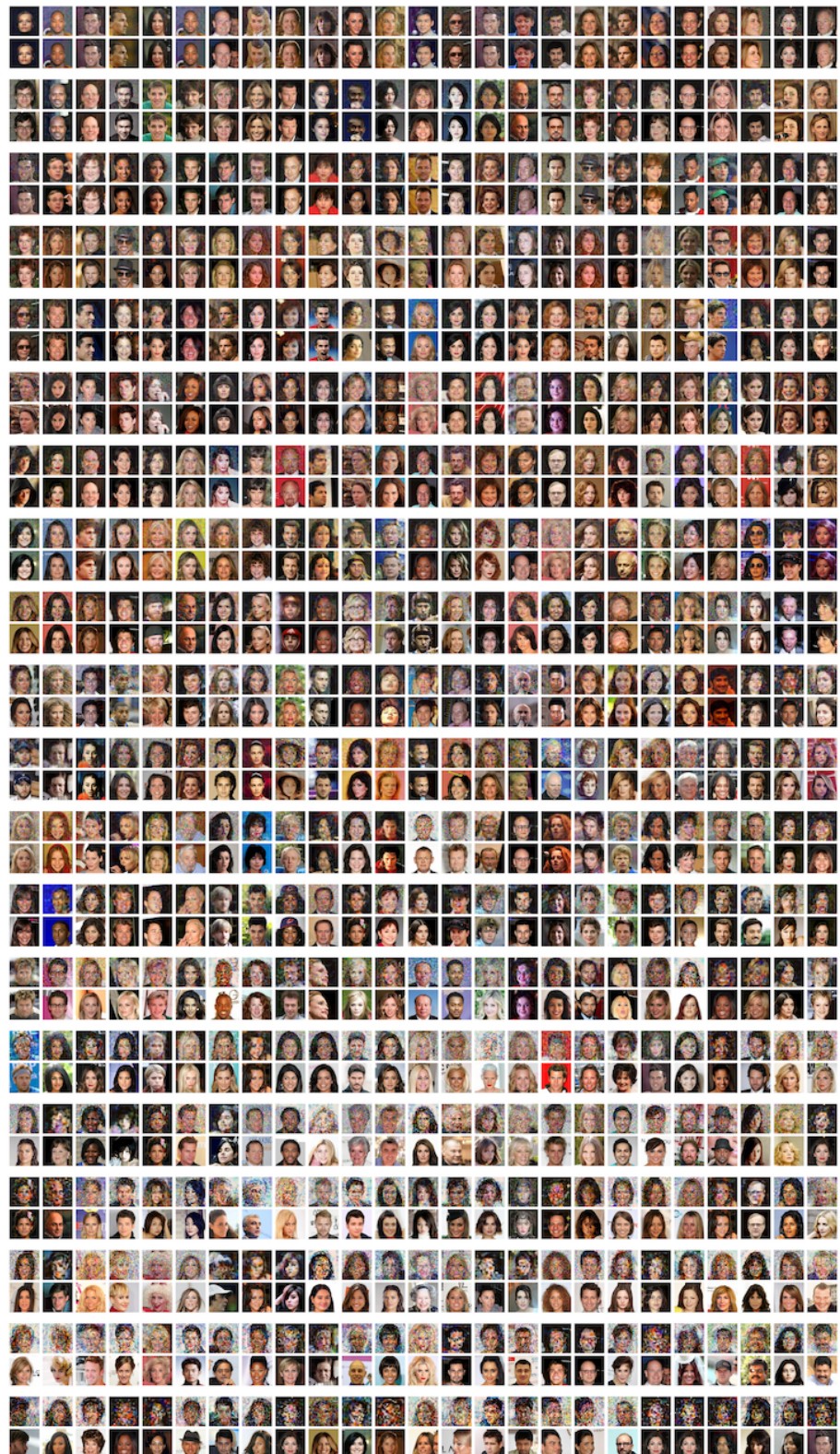

Figure 10: Reconstructions using a VAE from an RBF kernel trained using kernel regression on celebA.

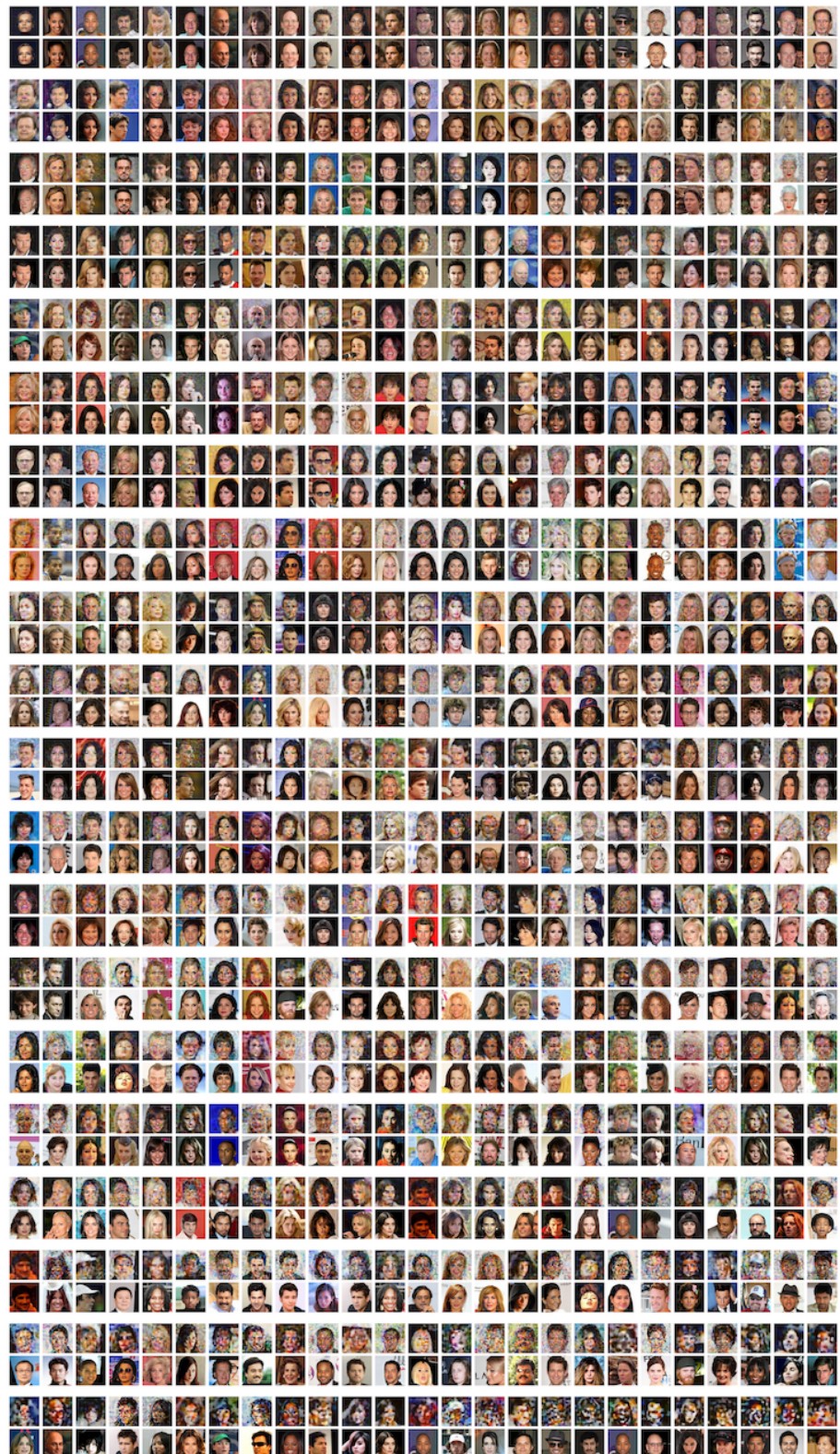

Figure 11: Reconstructions using a VAE from a Laplace kernel trained using kernel regression on celebA.

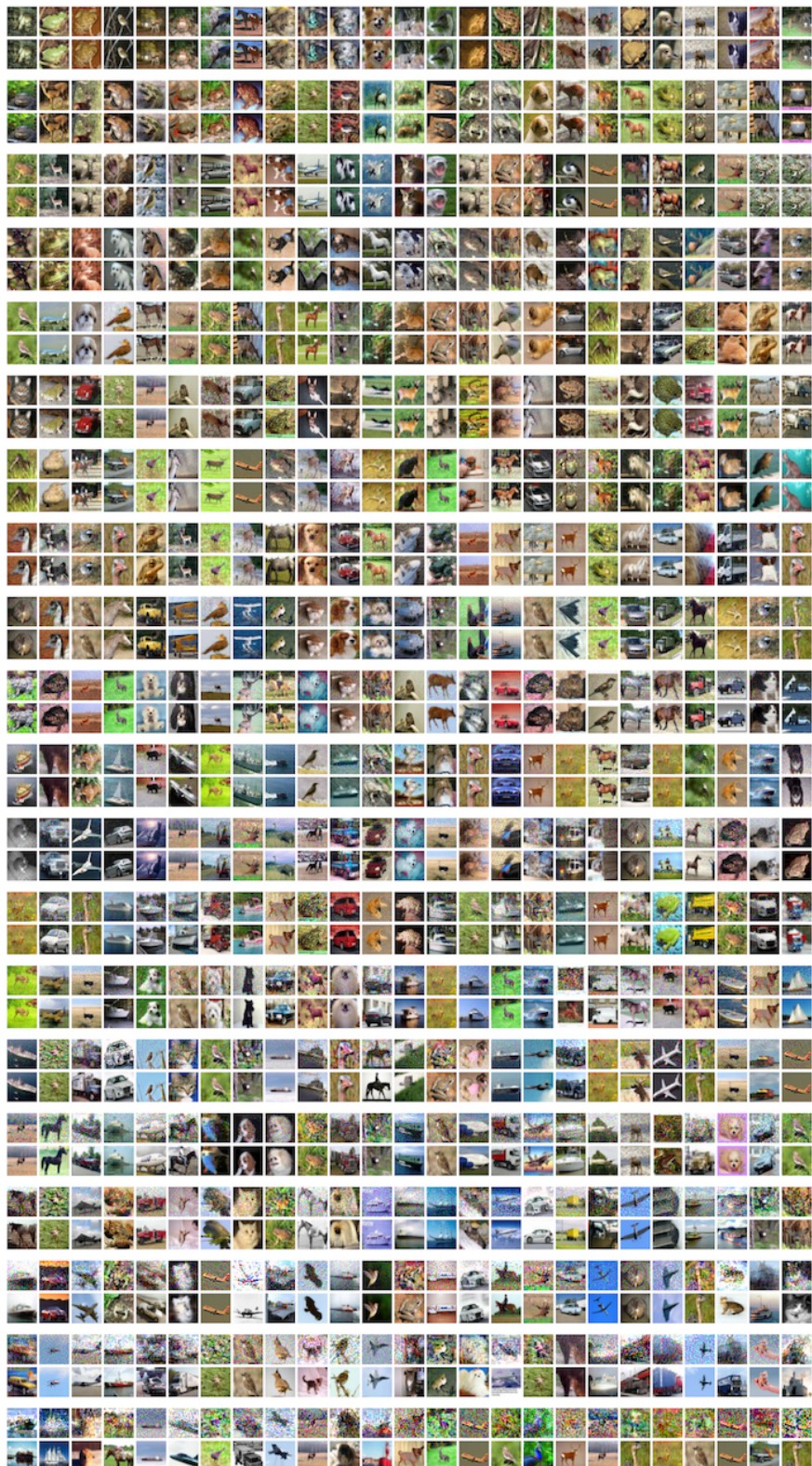

Figure 12: Reconstructions from an RBF kernel trained using kernel regression on CIFAR10.

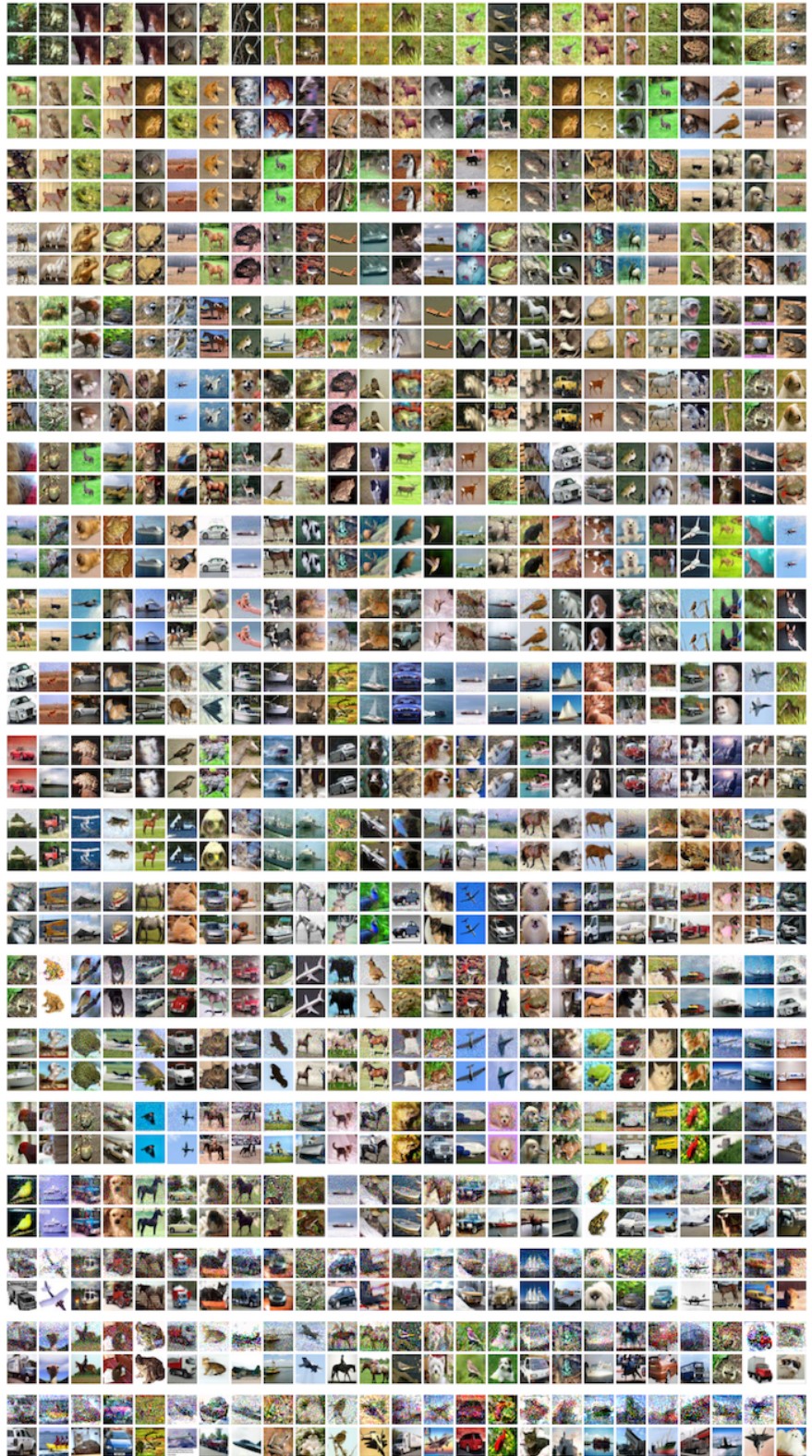

Figure 13: Reconstructions from a Laplace kernel trained using kernel regression on CIFAR10.

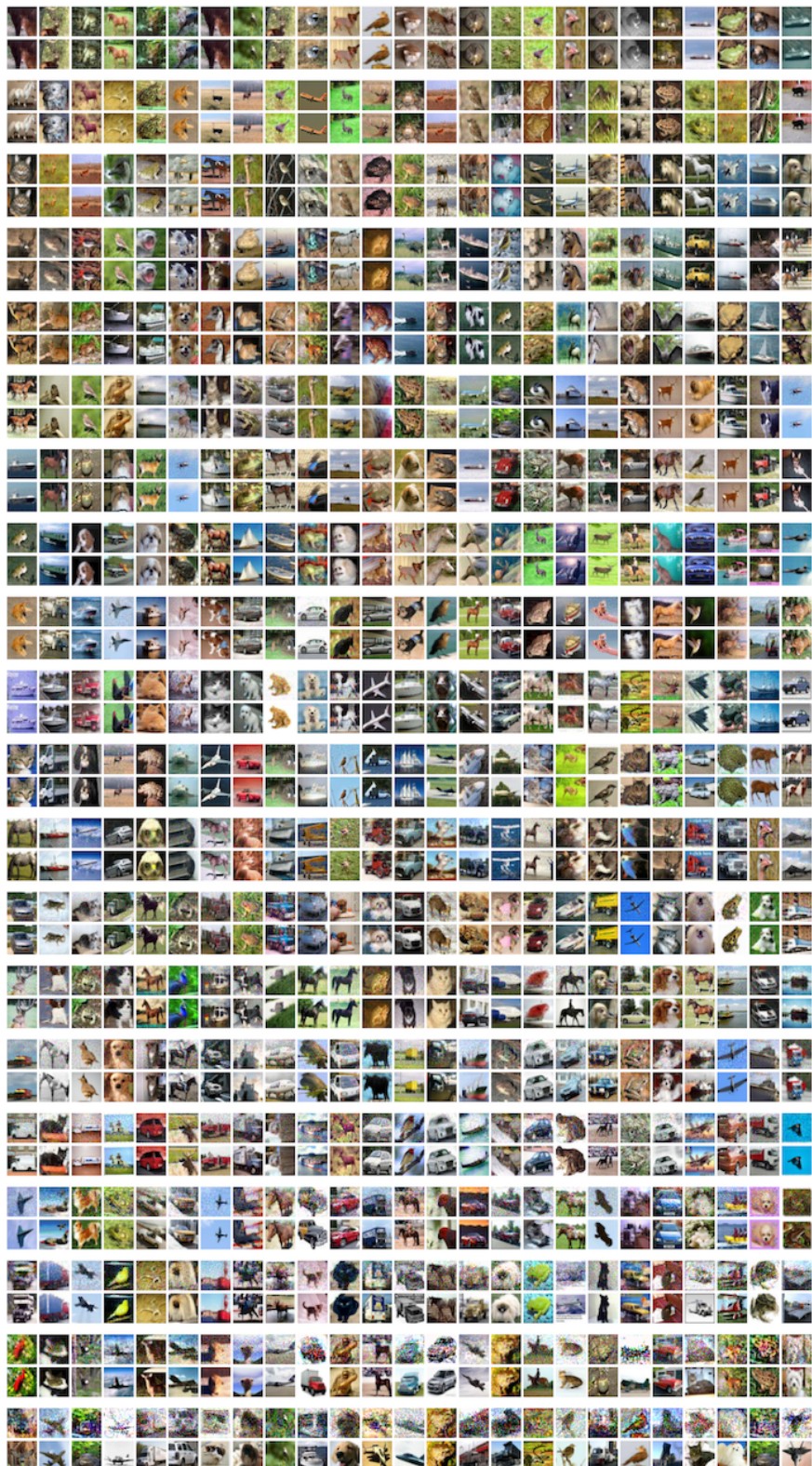

Figure 14: Reconstructions from a Laplace kernel trained using kernel regression with regularization $\lambda = 10^{-3}$ on CIFAR10.

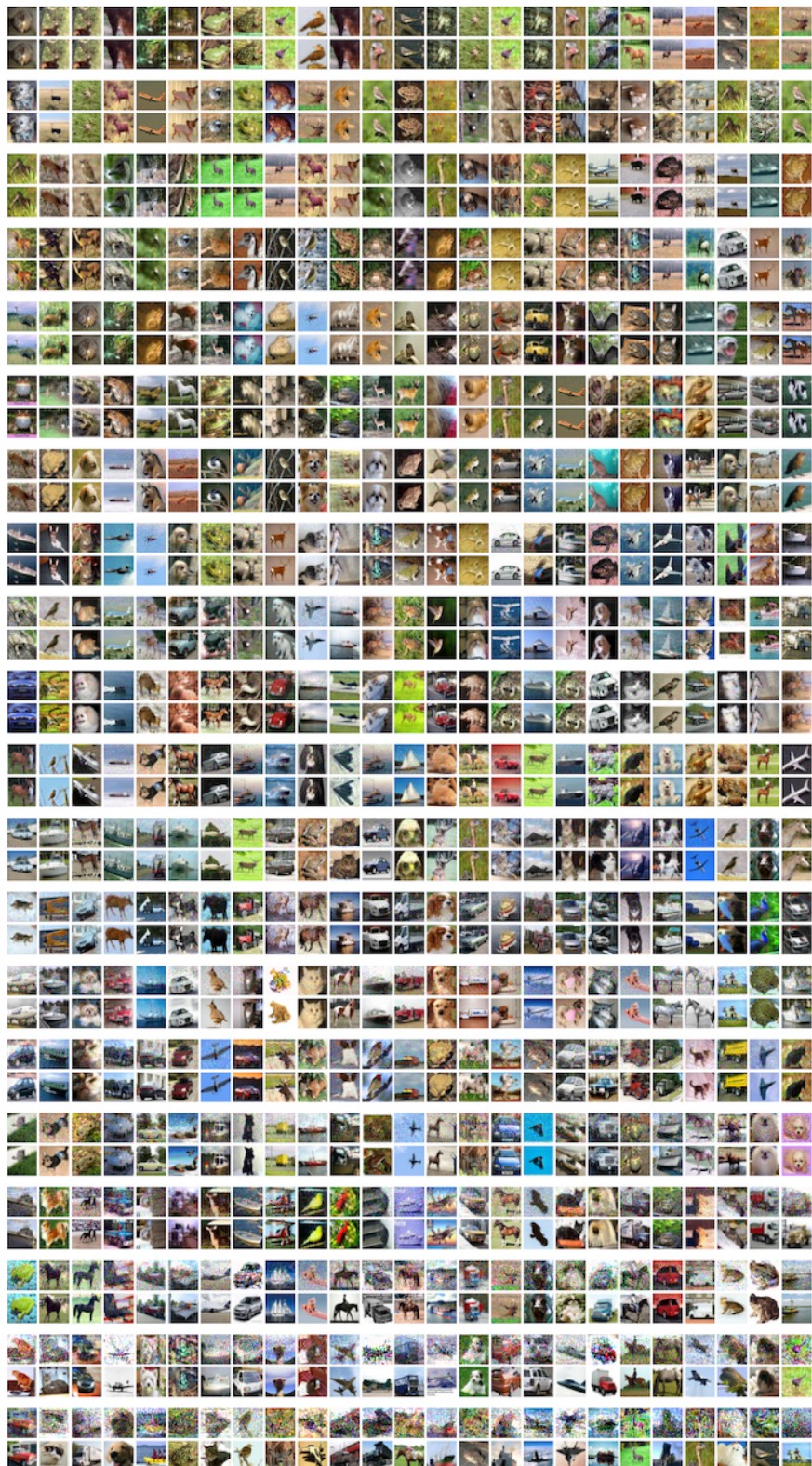

Figure 15: Reconstructions from a Laplace kernel trained using kernel regression with regularization $\lambda = 10^{-5}$ on CIFAR10.

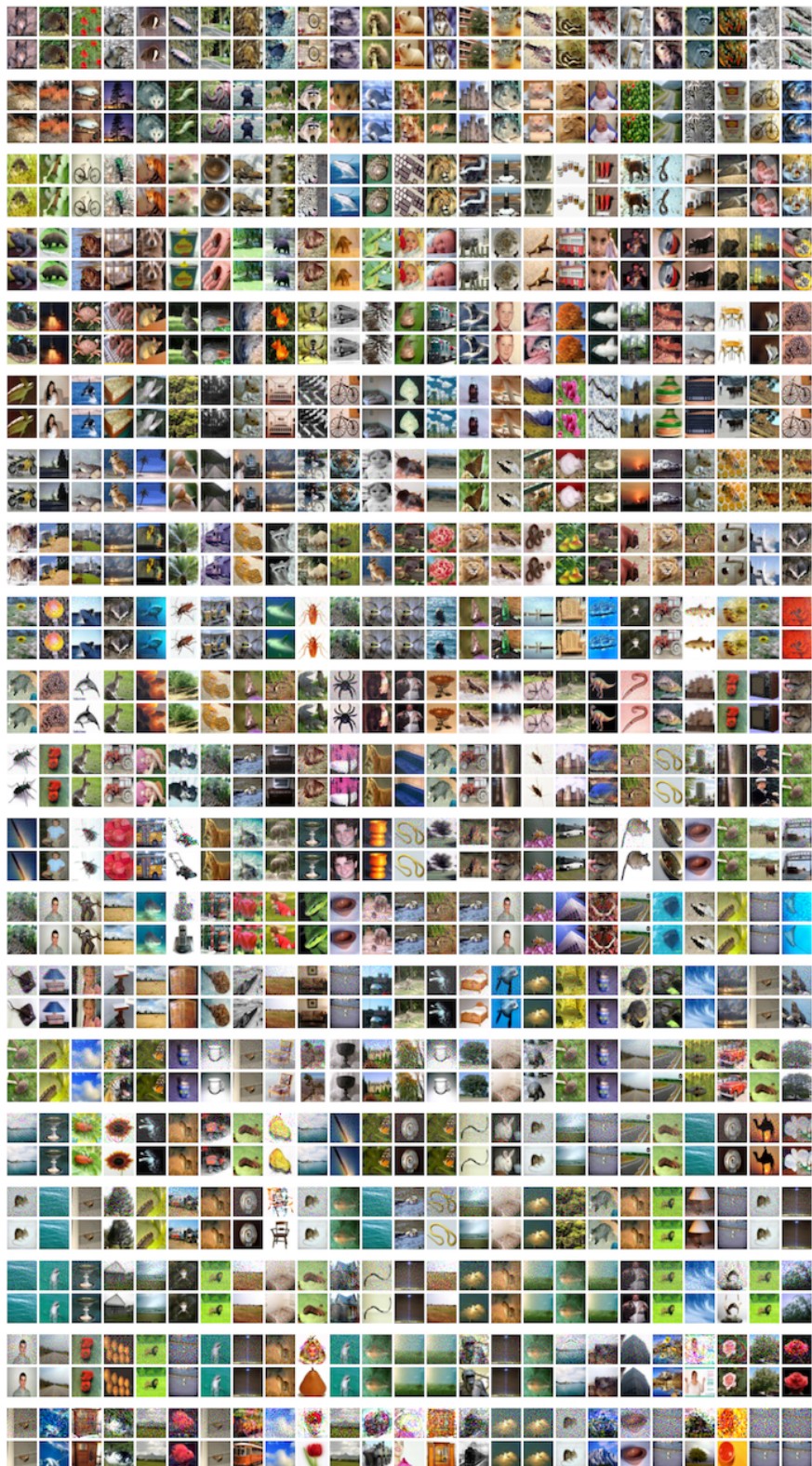

Figure 16: Reconstructions from an RBF kernel trained using kernel regression on CIFAR100.

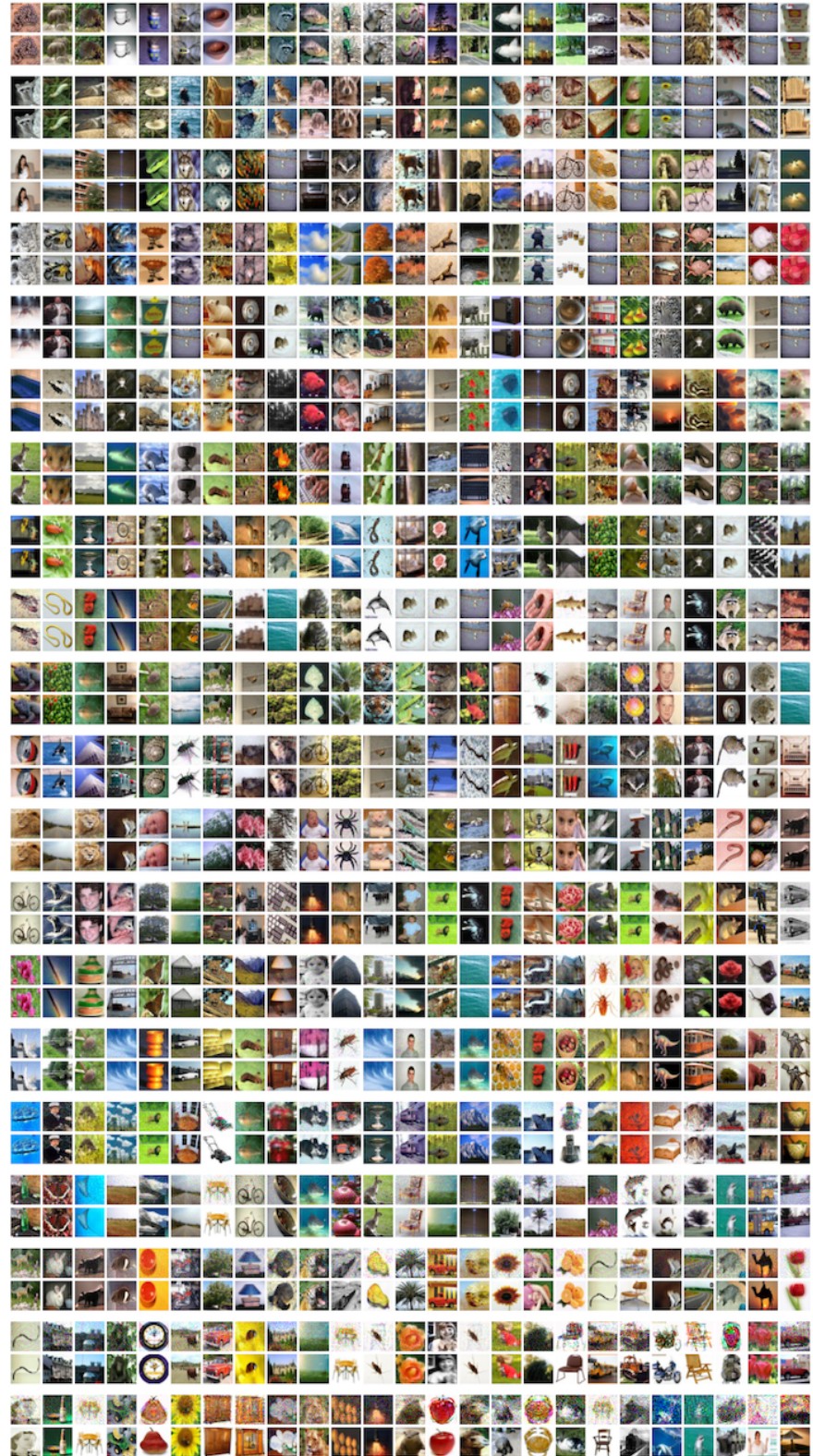

Figure 17: Reconstructions from a Laplace kernel trained using kernel regression on CIFAR100.

## E.2 SVM

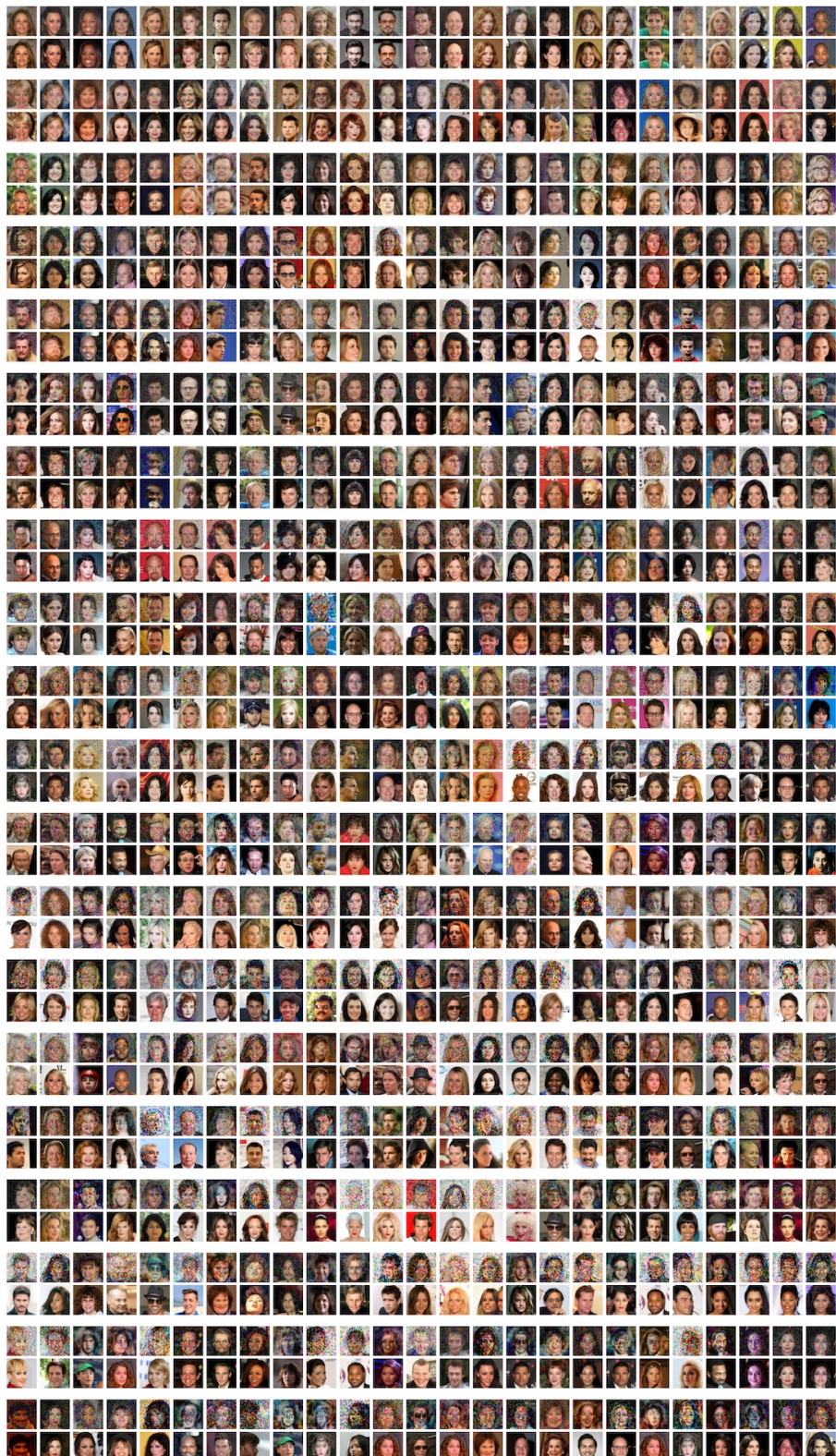

Figure 18: Reconstructions from an RBF kernel SVM on celebA.

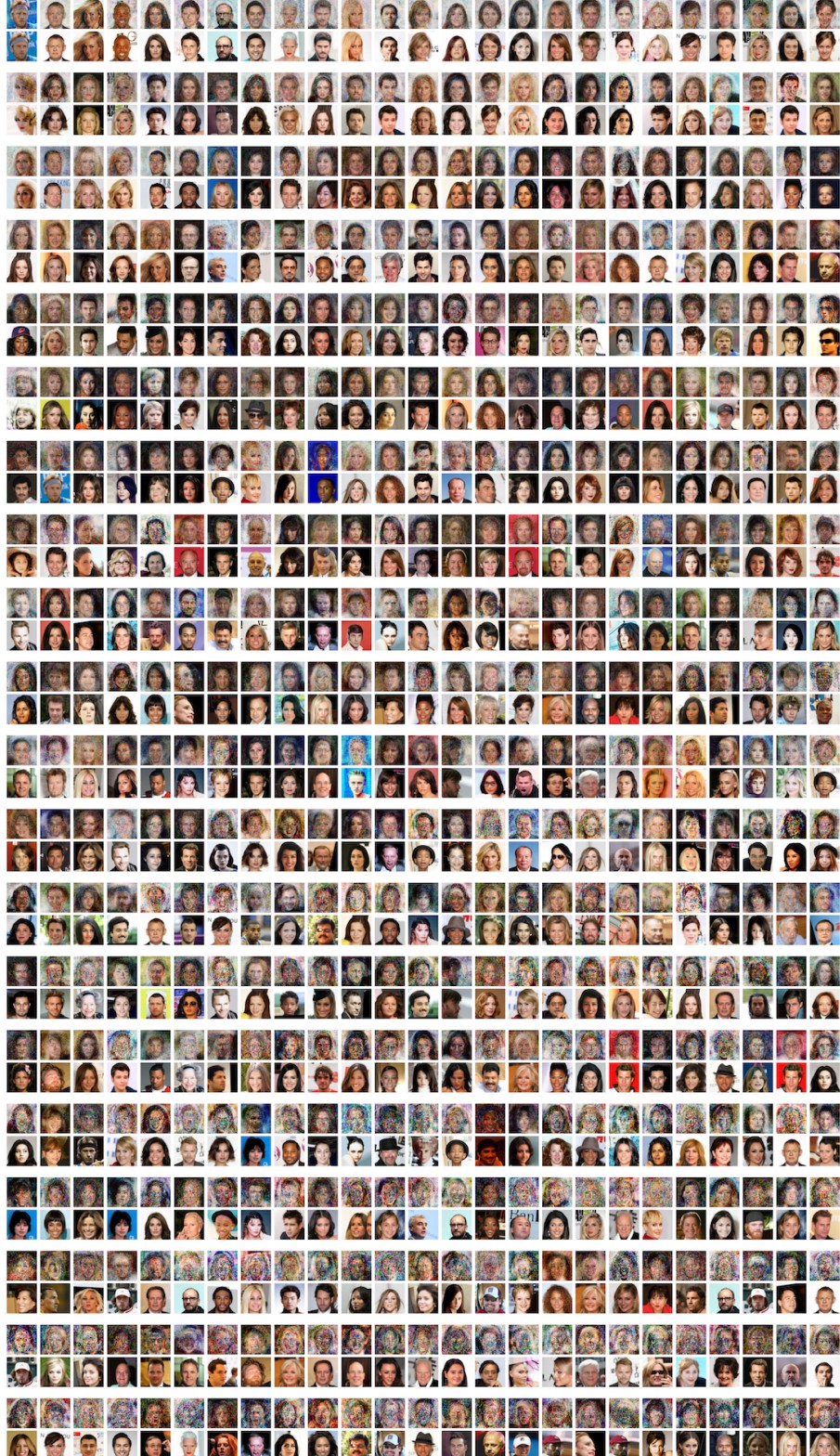

Figure 19: Reconstructions from a Laplace kernel SVM on celebA.

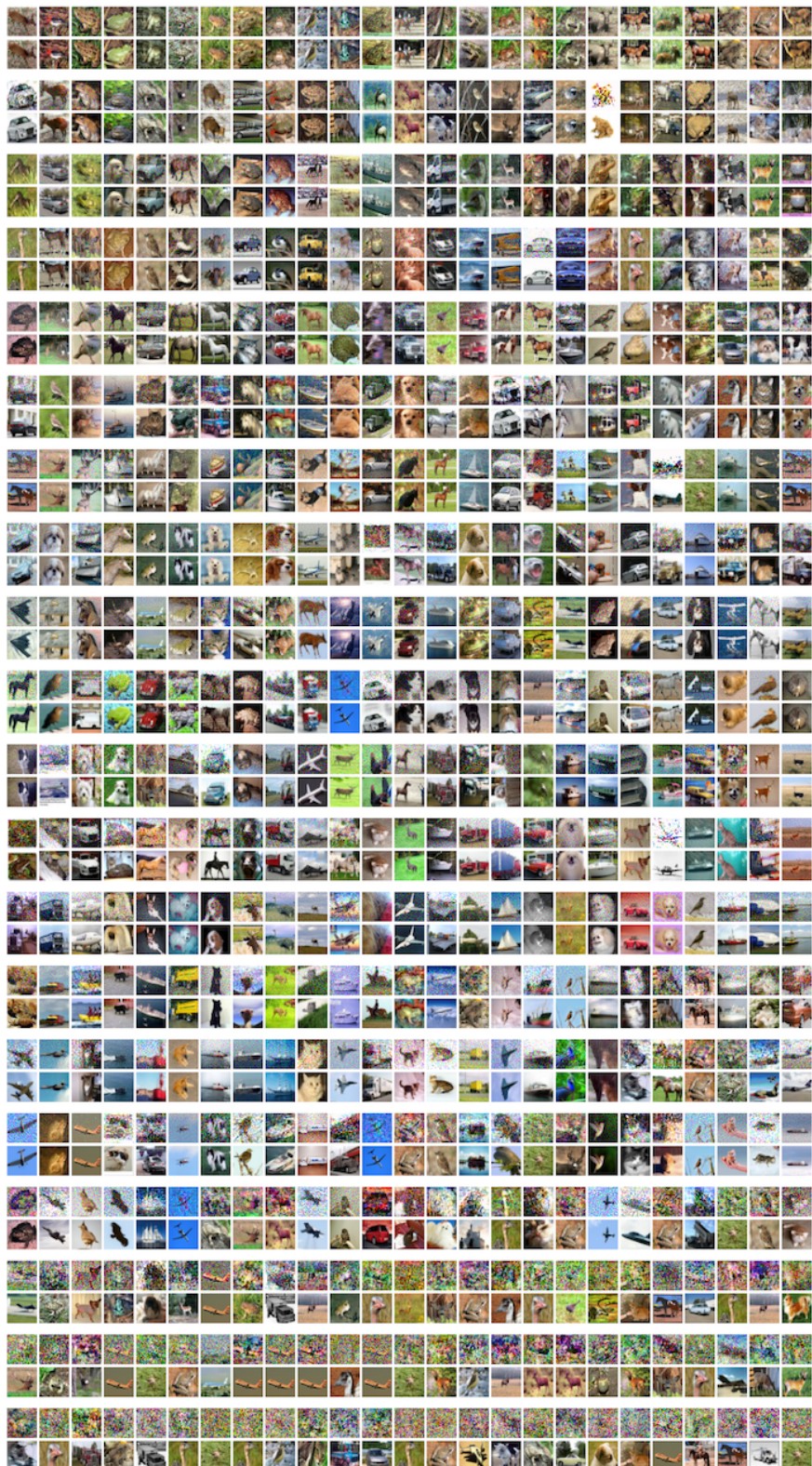

Figure 20: Reconstructions from an RBF kernel SVM on CIFAR10.

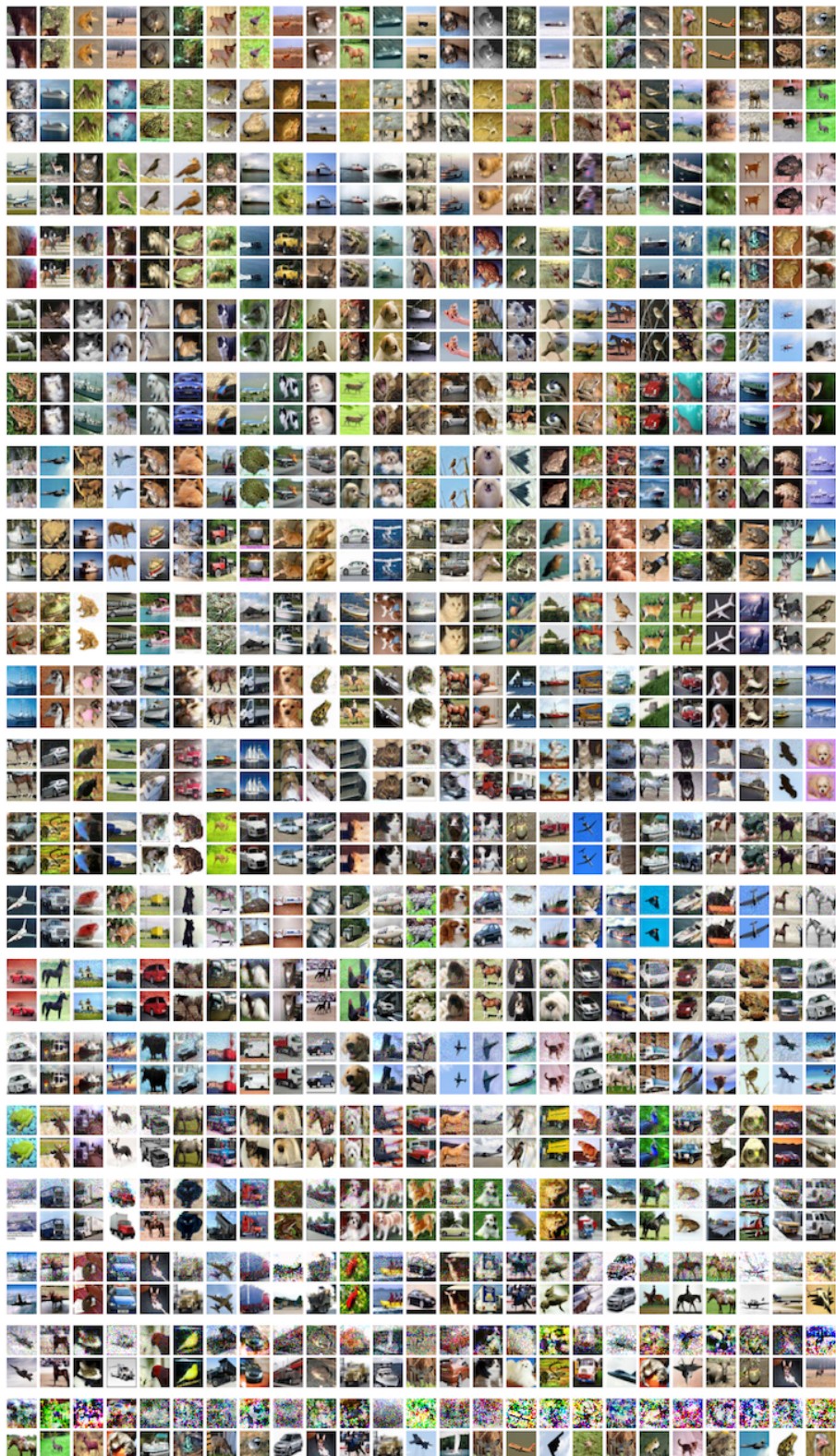

Figure 21: Reconstructions from a Laplace kernel SVM on CIFAR10.

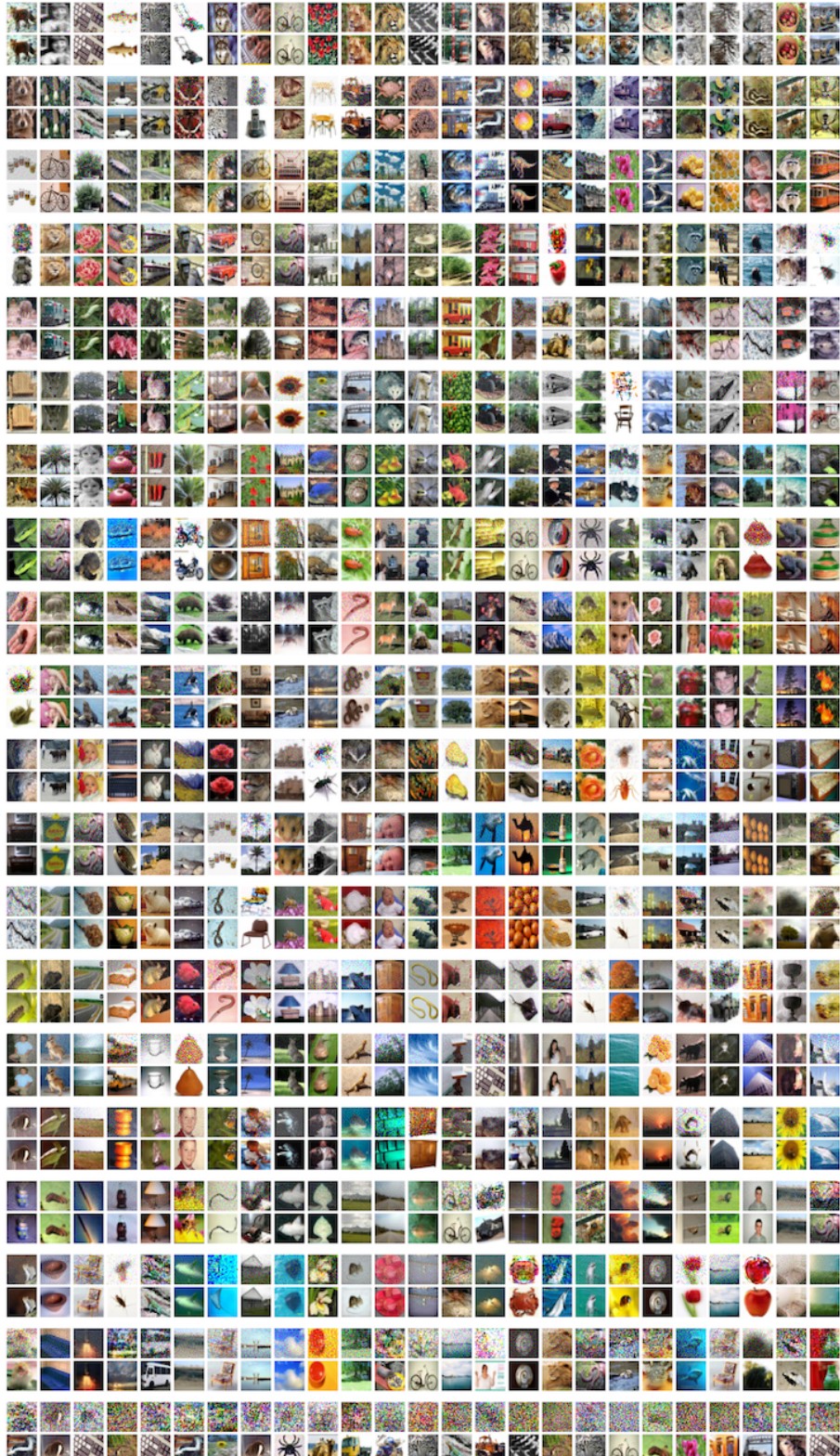

Figure 22: Reconstructions from an RBF kernel SVM on CIFAR100.

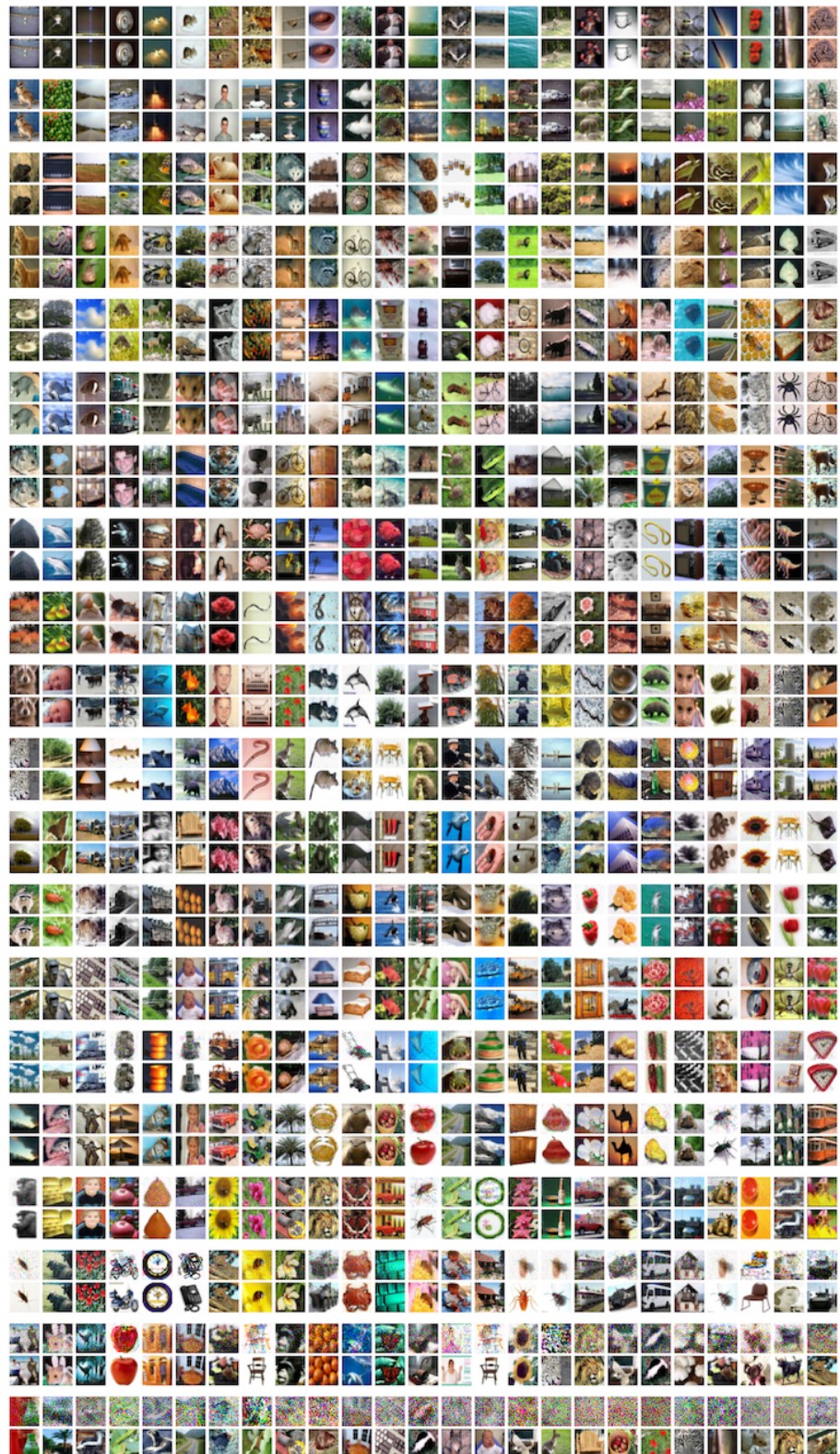

Figure 23: Reconstructions from a Laplace kernel SVM on CIFAR100.

