# OpenReview forum: "Querying Kernel Methods Suffices for Reconstructing their Training Data"
_TMLR — Accepted by TMLR_

### Review · Reviewer_5rAN · 2026-02-10

**Summary Of Contributions:**

**Summary of contributions**

This paper presents the theoretical analysis of reconstruction attacks on kernel methods (e.g., SVM) without access to model parameters. They support the theory by empirical results demonstrating the efficacy of their approach. The attack takes advantage of a kernel's ability to closely fit to the training data.

**Strengths**

- Kernels are analytically tractable making it easy to study the effect of memorization in a query-only access setting.
- The attack applies to a wide range of kernels in RKHS (reproducible kernel Hilbert space) represented by Eq (2).
- The attack assumes no knowledge of trained model's parameters and training data distribution ($D$). Furthermore, no labeled query points are necessary to run the attack. The modality of $D$ has be same as the training data, but simulated data can also suffice.
- The attack is shown to be robust to the choice of kernel used to approximate the attacked model conditioned on the assumption that the attacked kernel and the approximate kernel belong to the same family.
- Does not rely on computation in the feature space, since kernel operations can be done using feature mappings (owing to the kernel trick).

**Weaknesses**

- Theorems 2 and 6 constrain the attack to analytic and strictly p.d. kernels, limiting its general applicability to broader kernel-learning settings.
- The attack success is strongly affected by the number of training points (n) and the dimensionality of the input feature space (d). The query points have to be $ > nd/C + \<constant\>$.
- It is not mentioned by the authors, but kernel hyperparameters are assumed to be known by the attacker. For example, the $\gamma$ in the Laplace/RBF kernels. Assuming the adversary knows the type of kernel used in the attacked model, does the lack of access to kernel hyperparameters affect the efficacy of reconstruction?
- From Figure 4, it is clear that the proportion of training data reconstructed with DSSIM $<= 0.3$ is higher for Laplace/ RBF kernels. The attack seems already deteriorating when the kernel complexity grows. For example, Table 2 shows that with NTK, the efficacy drop to $23.2$% with quality reconstruction being just $4.6$%. But the authors do not comment on the possible cause(s) of this decline in performance.
 - *(Minor)* While the authors provide ample evidence of the attack's success (DSSIM < 0.3) for image datasets, the paper lacks evidence that this extends to other modalities. Another thing, for tabular datasets, can the authors comment on whether their method can be used for partial data reconstruction to impute missing/ unknown attributes?

**Audience:**

Yes

**Audience Explanation:**

The paper is an attempt at a detailed study of the feasibility of running query-only reconstruction attacks on kernel-based training methods. However, as highlighted in the **Weaknesses**, it leaves open to interpretation some of the results used to demonstrate the efficacy of the proposed attack.

**Claims And Evidence:**

No

**Claims Explanation:**

As highlighted above, the attack applies to multiple methods relying on kernels for learning, but the authors do not comment on the following:

- **Q1.** Why does the attack performance worsen as kernel complexity grows (such as for the polynomial and NTK kernels)?
- **Q2** Is the attack sensitive to the choice/ knowledge of kernel hyperparameters available to the attacker?
- **Q3** *(Minor)* The examples in the paper heavily rely on reconstructing data with image modality. Does the attack also work for other datasets, such as text or tabular data?

The lack of information in the paper addressing these questions affects the claims made by the authors.

**Requested Changes:**

I request the authors to address the **Weaknesses** and amend the paper accordingly (if necessary).

---

> ### Author Response · Authors · 2026-04-09
> **Response to Reviewer 5rAN**
>
> We thank the reviewer for their time. Below, we address the reviewer's concerns:
>
> - Theoretical assumptions: We agree that the theorems do not cover every possible kernel-learning setup. Our intent was to state assumptions that are mild enough to include some of the most common kernels in the literature, such as Laplace and RBF. Regarding strictly p.d., kernels used in practice tend to have strictly positive definite kernel matrices, as many algorithms break if the kernel matrix is not well-conditioned. The purpose of “almost analytic” is only to exclude pathological cases where equality on infinitely many sampled points would not determine the function globally. We will clarify this scope in the revision.
>
> - Number of query points: Indeed, the attack requires a number of query points that scales with $nd/C$. However, there is no parameter space dependence, ensuring the attack is still relatively efficient. All experiments in this paper were performed on a single GPU.
>
> - This is a good point, and the short answer is that the lack of exact hyperparameter knowledge does not prevent the attack. Section 6 already contains an experiment where the attacked kernel is only known to lie in the family $\exp(-\gamma \| x - x' \|^\beta)$ where both $\gamma$ and $\beta$ are unknown parameters that are optimized together with the other reconstruction variables. In that setting, the recovered parameters converge near the attacked ones and the reconstruction quality remains high. We refer the reviewer to Table 4 for the full results, and point out that for the Laplace Kernel, 82.4% of the dataset was reconstructed with DSSIM < 0.3 in both a KRR and SVM settings. We will make this more prominent in the main text.
>
> - Different kernels: Indeed, some kernels appear to be easier to attack than others. The theoretical results suggest that this is not a property of minima of the reconstruction loss, but likely arises in differences in the loss landscape and the difficulty in finding a minimum. We will indeed add a discussion in the paper. Thank you for this suggestion.
>
> - Tabular data: We agree that tabular datasets are an interesting modality for future work. We chose images as it is qualitatively easy to judge what a good reconstruction is. and aligned with related works.

---

> > ### Comment · Reviewer_5rAN · 2026-04-13
> > **Follow-up To Authors' Response**
> >
> > I appreciate the authors' detailed response. However, I would like to request the authors to provide us with a revised version of the manuscript with the promised adjustments/ clarifications. The Discussion section is not expanded with all the changes the authors commit to in their response to me (and the other reviewers). I cannot make a sound judgement about the quality of paper without the revised manuscript.

---

> > > ### Author Response · Authors · 2026-04-15
> > > **Follow-up**
> > >
> > > Thank you for your response. We have uploaded a revised version of the manuscript, with edits marked in blue. Please see the comment above titled "Revision Summary" for a list of the changes.

---

### Review · Reviewer_cBio · 2026-02-20

**Summary Of Contributions:**

This paper investigates whether training data can be reconstructed from kernel methods using only query access to the model (i.e., without access to model parameters). The authors make three main contributions: 1) A query-based reconstruction attack applicable to various kernel methods including kernel ridge regression, SVM, and kernel density estimation. 2) Theoretical guarantees showing that for strictly positive definite and "almost analytic" kernels, minimizing the reconstruction loss with sufficient query points (m > n(d+2)) guarantees exact recovery of the training data with probability 1 over randomly sampled queries.

Strengths
+ Provides rigorous proof that reconstruction is possible with query-only access under mild conditions

+ Demonstrates that hiding model parameters is insufficient protection against data reconstruction

+ Tests multiple kernels, tasks, and datasets with clear metrics

Weaknesses

- Limited dataset scale: Evaluation only on CIFAR-10/100 (32×32) and celebA (64×64); no ImageNet-scale validation

- Limited evaluation metrics: Relies primarily on SSIM/DSSIM; lacks other perceptual metrics (LPIPS, PSNR, FID) that could provide complementary insights

- Assumes kernel knowledge: Attack requires knowing the kernel family used (though authors show some robustness to hyperparameter uncertainty)

- Query requirements: Theoretical bound requires many queries (n(d+2)), though empirically fewer suffice

- Limited to kernel methods: Results don't directly extend to neural networks; connection to prior parameter-access attacks (Haim et al.) could be better contextualized

**Audience:**

Yes

**Audience Explanation:**

The paper targets data privacy, which is relevant to TMLR.

**Claims And Evidence:**

No

**Claims Explanation:**

- Dataset scale: All experiments use small images (32×32, 64×64, 128×128 with VAE). The claim that this attack poses a "major security concern" (page 1) for real-world applications remains unsubstantiated without ImageNet-scale validation. The attack may not scale to higher resolutions where the dimension d is much larger, dramatically increasing query requirements.

- Metric limitations: Relying primarily on SSIM/DSSIM is problematic. SSIM captures structural similarity but may not reflect semantic reconstruction quality. Including LPIPS (which better correlates with human perception) or classification accuracy on reconstructed images would strengthen the evidence that true training data (not just visually similar samples) is recovered.

**Requested Changes:**

1. Better contextualize relative to prior work: The paper currently claims "there are no works to directly compare against" (page 2), but Haim et al. (NeurIPS'22) demonstrated optimal reconstruction on well-trained models. While Haim et al. requires parameter access and this work is query-only, the framing should be adjusted to:
- Acknowledge that parameter-access attacks have shown strong results on well-trained models

- Clarify that the key novelty is achieving comparable or better results without parameter access

- Discuss whether the theoretical optimal reconstruction bounds from Haim et al. apply in the query-only setting

- Explain how the query-only constraint changes the reconstruction problem fundamentally

2. Add baselines for reconstruction quality when training data is held-out: Current comparison only shows that reconstructions match training points, but doesn't demonstrate that the method doesn't simply generate plausible samples that happen to match training data. Include experiments showing that reconstructions are significantly closer to training points than to held-out test points.

3. Validate on larger-scale datasets: The current evaluation is limited to CIFAR-10/100 (32×32 images) and celebA (64×64). To establish broader applicability, experiments on ImageNet-scale data (224×224 images, 1000+ classes) would be valuable. This would test whether the attack scales to higher dimensions and more complex data distributions.

4. Expand evaluation metrics beyond SSIM/DSSIM: The paper relies primarily on SSIM-based metrics (DSSIM). Include additional perceptual metrics such as LPIPS, PSNR, and FID to provide a more comprehensive assessment of reconstruction quality.

---

> ### Author Response · Authors · 2026-04-09
> **Response to Reviewer cBio**
>
> We thank the reviewer for their time. Below, we address the reviewer's concerns:
>
> - Scale: The concerns regarding the scale are fair, but should be viewed together with a few points. First, the scale of experiments actually exceeds that of the closest methods in the literature that we compare ourselves to (Haim et al., 2022, Loo et al., 2023., Buzaglo et al., 2024). Second, kernel methods are not used at imagenet scales, as the cost of training has a poor dependence on the number of samples. In contrast, they are known to either match or even outperform neural networks for smaller datasets, which is where our paper focuses. Third, the experiments in this paper ran on a single GPU, whereas large labs have access to compute that is orders of magnitude greater. Lastly, the attack complexity is independent of parameter count, unlike in parameter-based reconstruction attacks. We will add a discussion in the paper to clarify these points.
>
> - Metrics: Indeed, metrics are an issue in reconstruction works, but this is not specific to our paper. Previous papers also used L2 and SSIM based metrics instead of something like LPIPS, as reconstructed images fall slightly outside the distribution of natural images (e.g. there is often some "noise"). As such, feature based approaches such as LPIPS are known to be less meaningful in these settings. Additionally, PSNR which you referenced is just a simple function of the L2 error, which we already report. We will gladly add a short discussion of this in the paper
>
> - Kernel family knowledge: In practice there is a small number of kernels that are frequently used, so the atttacker can easily try them all and identify the right one (e.g., by monitoring the loss). Together with our results in Table 4 that demonstrate successful reconstruction without knowledge of kernel hyperparameters, we believe that this is not a serious limitation. We will add a discussion on this in the paper.
>
> - Query requirements: It is true that in practice, slightly fewer queries suffice than the worst-case bounds in our theorem. Nevertheless, our theory does put us in the correct ballpark in terms of the number of query points needed, as demonstrated in Figure 4 (a).
>
> - Kernels methods: We acknowledge that our reconstruction attack is specifically for kernel methods and not neural networks. We kindly ask the reviewer to consider that kernel methods remain a valid and active area of research that interests many people in the community.
>
> - Connection to Haim et al: We will happily expand the discussion on this paper. In fact, we were in discussions with the authors of Haim et al. continuously throughout our work, and they were very supportive and excited by our results. We do of course acknowledge that parameter-access attacks showed strong results, and we will make sure this comes across in the paper. Moreover, we will gladly elaborate on how the query-only constraint changes the problem. The KKT-based approach in Haim et al. is fundamentally designed for parameter-based attacks, and does not directly carry over.
>
> - "Optimal Bounds of Haim et al": We note that Haim et al. did not prove any theoretical bounds, and they did not claim that their reconstruction attack is optimal.

---

### Review · Reviewer_ZgCm · 2026-03-29

**Summary Of Contributions:**

This paper investigates data reconstruction attacks against kernel-based predictors. Specifically, the proposed attack jointly optimizes the dual coefficients of the kernel and a set of pseudo-data points (intended to mimic the original training data) by minimizing the discrepancy between: (1) the predictions of a surrogate kernel model built from the current dual coefficients and pseudo-data, and (2) the query outputs obtained from the target (i.e., the "attacked") model on a set of inputs drawn from the same distribution as the training data.

Experimental results on the CIFAR-10 and CelebA datasets demonstrate visually plausible reconstructed samples. Quantitative evaluation using DSSIM scores further supports the effectiveness of the proposed reconstruction attack.

**Audience:**

Yes

**Audience Explanation:**

Yes, to some extent. There appears to be limited in-depth investigation of data reconstruction attacks specifically targeting kernel methods in prior work, so the topic itself could be of interest to TMLR’s audience—particularly if the paper were to provide clear, substantive new insights.

However, in its current form, the contribution seems limited. The main takeaway largely aligns with a well-understood intuition: that non-parametric methods, such as kernel-based predictors built directly on training data points, can be vulnerable to privacy attacks. Without deeper analysis, stronger empirical evidence, or more novel insights, the overall level of interest may remain moderate.

**Broader Impact Concerns:**

No serious negative consequences seem to arise directly from this work in its current form. There is a broader impact section included in the paper; however, it appears somewhat brief and could be further enriched with a more detailed discussion.

**Claims And Evidence:**

No

**Claims Explanation:**

I evaluate this work along the following key dimensions: the novelty and significance of the problem, the proposed methodology, the insights and practical relevance of the reported results, and the overall presentation quality.

Overall, I am concerned that the paper largely resembles a re-implementation of prior work. The problem itself has already been explored in the literature (e.g., Tramer et al., 2016), and the proposed method appears to offer limited conceptual novelty. In particular, my understanding is that Section 4.1.3 of Tramer et al. (2016) already describes the attack procedure adopted in this submission; please clarify if there are substantive differences. Furthermore, the paper does not seem to provide significant new experimental insights or practically meaningful takeaways. As a result, the work gives the impression of a proof-of-concept rather than a thorough and substantive advancement of the state of the art. The investigation also appears somewhat incomplete in its current form, and the presentation could be improved to enhance clarity and rigor.

- For instance, the experimental evaluation is limited to relatively simple image datasets such as CIFAR-10 and CelebA. While this is acceptable for a proof-of-concept study, it does not adequately reflect realistic or high-stakes deployment scenarios in modern machine learning systems. Moreover, the reported results are not entirely convincing in demonstrating that the proposed attack constitutes a serious practical threat, particularly given the strength of the underlying assumptions and the lack of broader empirical validation.

- The threat model is not clearly defined. Although the paper prominently claims that only query access to the target model is required, the method appears to rely on additional knowledge—such as the choice of kernel function and possibly its hyperparameters (e.g., the regularization parameter ( \lambda )). These assumptions should be explicitly stated and their practicality carefully justified. It would further strengthen the work to discuss whether, and to what extent, these assumptions can be relaxed. While some related discussion is included (e.g., in Section 6), a dedicated and well-structured threat model section would be more appropriate, where each assumption is clearly enumerated and directly linked to corresponding discussions or empirical evaluations.

- Algorithm 1 is not fully self-contained.
  1. Referring to $n$ as “reconstruction points” is somewhat ambiguous; it would be clearer to explicitly define it as the number of data points to be reconstructed. This also raises an important question regarding the threat model: does the attack assume prior knowledge of the training set size (or, more generally, its local density around query points)? If so, this assumption should be clearly stated and justified, and the paper should discuss how sensitive the results are to this knowledge.

  2. Line 5 states “Update $\hat{x}_i$, $\hat{\alpha}_i$" via an optimization step,” but the optimization procedure is not clearly specified.  I expected a more formal and detailed description in Algorithm 1, as well as clarification in Section 4.1. However, Section 4.1 currently only present the target training algorithms and does not sufficiently describe how the proposed attack is actually implemented.

- If the analytical results and proofs are intended to be a key contribution of the paper, they would benefit from clearer structuring and stronger motivation. In the current form, it is not entirely clear what concrete insights the theoretical results provide about the reconstruction attack, and several parts of the proofs are difficult to follow.
   -  Some of the results appear to be relatively straightforward consequences of standard assumptions. For example, Proposition 3 seems to follow directly from the strictly positive definite property of the kernel, yet this assumption itself is not thoroughly justified or discussed in terms of when it holds in practice.
   - The bounds related to the required number of query samples seems may be possible to be developed in greater depth. In their current form, they appear to largely reflect a basic counting argument (one needs at least as many independent equations as unknowns when viewing the reconstruction as solving the linear system involving the kernel matrix). The presentation would benefit from a more rigorous treatment and a clearer explanation of what new insight these bounds provide beyond this baseline perspective.

**Requested Changes:**

- Clarify the novelty and positioning relative to prior work (e.g., Tramer et al., 2016). Explicitly state what is in principle new in the proposed method and how it differs from existing attack procedures, and strengthen the paper with clearer, practically meaningful insights beyond a proof-of-concept.

- Strengthen the experimental evaluation by going beyond simple datasets such as CIFAR-10 and CelebA, or by better justifying their relevance. Provide stronger empirical evidence that the attack constitutes a realistic and significant threat under practical settings.

- Introduce a clearly defined threat model section. Explicitly enumerate all assumptions (e.g., knowledge of the kernel function, hyperparameters, potential knowledge of training set size), justify their practicality, and discuss which assumptions can be relaxed, with pointers to supporting analysis or experiments.

- Improve the clarity and completeness of Algorithm 1.  Provide a precise and self-contained description of the optimization procedure (joint vs. alternating updates, optimization method, etc.), and ensure consistency between the algorithm and its description in Section 4.1.

- Refine the theoretical analysis and proofs, e.g., better motivate the role and implications of each result for the reconstruction attack; Clarify assumptions (e.g., strictly positive definite kernels) and discuss their practical validity.

- Provide more insight and discussion on the attack method. For example, it is unclear what mechanisms (if any) prevent the reconstructed points from collapsing to a small number of modes beyond random initialization; it would be helpful to clarify whether such mode collapse is a concern in practice. In addition, the scalability with respect to n is not sufficiently explored: further analysis of how sensitive performance is to the choice of $n$, and how the method behaves as n increases, would strengthen the paper.

- Better contextualize the evaluation metric (DSSIM). It would be helpful to provide reference values to aid interpretation: for example, what DSSIM range corresponds to visually similar images? How do the reported values compare to those between random image pairs or nearest neighbors within the same class in datasets such as CIFAR-10 and CelebA? Including such baselines would make the quantitative results more meaningful and easier to interpret.

---

> ### Author Response · Authors · 2026-04-09
> **Resonse to Reviewer ZgCm**
>
> We thank the reviewer for their time. Below we address the reviewer's concerns:
>
> - Relation to Tramer et al., 2016: While their results are certainly relevant, there are many important differences. We will elaborate on these in the paper. In particular:
>     - Theory: Among our results is a theorem that provides both theoretical support for our method, as well as an understanding of why such a query based reconstruction attack works. Such an understanding is absent from Tramer et al. 2016.
>     - Algorithmic/methodological differences: First, the attacks of Tramer et al. are based on "reimplementing the training procedure", in the sense that if, for example, the kernel was trained via logistic regression, then the reconstruction attack also uses the logistic loss. Our method does not require assumptions on the training procedure of the kernel. Second, we consider leveraging data following a similar distribution, such as synthetically generated images. Third, we consider optimizing for unknown kernel hyperparameters, which are generally unknown (see Table 4). Fourth, we consider combining our reconstruction attack with VAEs, allowing us to scale to 128x128 CelebA images. Lastly, there are many other minor implementation details as specified in Appendix A.
>      - Empirical differences: Their results were on a tiny scale, as they reconstructed only 20 MNIST-style images of resolution 14x14. The intrinsic complexity of this data type is very low, and it is well known that many methods that work on MNIST style data do not even translate to CIFAR10.
>
> - Experimental Scale: This is a fair concern, but it should be viewed together with a few points. First, the scale of experiments actually exceeds that of the closest methods in the literature that we compare ourselves to (Haim et al., 2022, Loo et al., 2023., Buzaglo et al., 2024). Second, kernel methods are not typically used at very large scales, as the cost of training scales poorly with the number of samples. In contrast, they often match or even outperform neural networks on smaller datasets, which is where our paper focuses. Third, the experiments in this paper ran on a single GPU, whereas large labs have access to compute that is orders of magnitude greater. Lastly, the attack complexity is independent of any parameter count. We will add a discussion in the paper to clarify these points.
>
> - Threat model: Thank you for this comment, we will clarify exactly what knowledge is required. We note that kernel hyperparameters can often be learned as we demonstrated in section 6 and Table 4.
>
> - The attack does not require exact knowledge of the training set size. As we demonstrate in Table 3, the attack works when the number of points to be reconstructed is both larger or smaller than the number of training points, with best results when it is larger. Therefore, if the number of training points is unknown, it is better to over-guess.
>
> - The attack is not tied to a specific optimizer. We used Adam and will mention this in the main part of the paper. We note that this is already discussed in Appendix A, where all the implementation details are fully specified. We believe this is the common practice, so that the algorithm box in the main paper is as readable as possible. We will also add a link to the GitHub code in the camera ready version.
>
> - Theoretical Results: The theoretical results are indeed a key contribution of the paper. Regarding the assumptions they are satisfied by many common kernels such as the Laplace and RBF kernel. Regarding the proofs, indeed Proposition 3 is not the main part of the argument, and is mainly used as a lemma for Theorem 2. However, we believe that the proof technique in Theorem 2 is novel in the related literature as it utilizes tools in smooth manifold theory (namely the Submersion Level Set Theorem) and the analyticity properties of the kernel. In short, the theorem proves that w.p. 1 model outputs on the queries uniquely define the model on the whole space using a manifold-dimension argument. Moreover, Theorem 2 handles kernels with a countable number of non-analytic points (such as the Laplace kernel) by adding additional queries and utilizing smoothness properties of the mapping between training points and non-analytic points.
> We will clarify the contributions in the paper.
>
> - DSSIM: In the past work of Haim et al. 2022, the threshold of DSSIM < 0.3 was established as interpreting two images as highly similar. Of course, this is up for interpretation, and there is no universally correct number, which is why we also include many qualitative results in the appendix. The reconstruction figures in Appendix E clearly confirm that images in the training set are being reconstructed.
>
> - No Mode Collapse: Collapsing to a small number of modes would not minimize the loss as our theory demonstrates. In practice, we occasionally observe a few duplicates, but this is rare, as can be seen in the reconstruction figures in Appendix E.

---

### Author Response · Authors · 2026-04-15
**Revision Summary (Addressed to all reviewers)**

Dear Reviewers,

We have uploaded a revised version of the manuscript following our discussion. The changed text is marked in blue. We summarize here the changes and the reviewers whose comments they address:

- An extended discussion of the relation of our work to Tramer et al. **[ZgCm]**
- Algorithm 1: Changed to "number of points to reconstruct", and referenced Appendix A, where all implementation details are specified. **[ZgCm]**
- A "Threat Model" Section (Sec 4.1) that makes very clear what knowledge/access the attacker needs, and what it doesn't, including hyperparameter and kernel knowledge **[5rAN, cBio]**.
- Clarifications in Section 5 regarding the assumptions of the theoretical results and their role/importance **[5rAN, ZgCm]**
- Minor edit of the discussion in Section 6 "Comparison Between Different Kernels." which explains why some are easier to reconstruct than others **[5rAN]**. We note that most of the discussion on differences in performance is not in blue, as it was present in our original submission.
- Extended the Discussion in Section 7 around the metrics used **[ZgCm, cBio]**.
- Better contextualized Haim et al in Section 7.1 **[cBio]**.

We appreciate your help in improving our paper.

---

### Decision · Action_Editor_F1K3 · 2026-05-25

**Recommendation:** Accept as is

**Audience:**

Yes

**Audience Explanation:**

As the paper discusses a plausible threat against machine learning models, I think it is interesting for the broad TMLR audience. All the reviewers agree with this as well.

**Claims And Evidence:**

Yes

**Claims Explanation:**

This paper studies reconstruction attacks against kernel-based classification methods. Authors make both theoretical and empirical claims about kernel-methods' susceptibility to reconstruction of the training data. Authors demonstrate on two image classification data sets that the proposed attack indeed reconstructs visually similar images to the training data, and also show quantitative results using similarity metrics. The proposed theoretical result is about establishing an upper bound for the number of query points needed for the reconstruction. While the Thm 2 based upper bound seems rather pessimistic, authors derive a more reasonable heuristic upper bound from it that seems to give reasonable empirical performance as well.

Two of the three reviewers were satisfied with the claims and evidence presented in the paper. The third reviewer was mainly concerned about the scope of the experiments, not so much if the current experiments support the claims. I with this reviewer, that more experiments with larger data sets would make the paper more convincing. However, I do also understand authors' reasoning on sticking with smaller data sets: that is the domain where kernel methods still find use over deep neural networks. Therefore, in my opinion the paper does justify its claims with enough evidence.